# DIVER: Diving Deeper into Distilled Data via Expressive Semantic Recovery

## Abstract

Dataset distillation aims to synthesize a compact proxy dataset that is unreadable or non-raw from the original dataset for privacy protection and highly efficient learning. However, previous approaches typically adopt a single-stage distillation paradigm, which suffers from learning specific patterns that overfit on a prior architecture, consequently suppressing the expression of semantics and leading to performance degradation across heterogeneous architectures. To address this issue, we propose a novel dual-stage distillation framework called **DIVER**, which leverages the pre-trained diffusion model to dive deeper into **DI**stilled data **V**ia **E**xpressive semantic **R**ecovery, a process of semantic inheritance, guidance, and fusion. Semantic inheritance distills high-level semantic knowledge of abstract distilled images into the latent space to filter out architecture-specific "noise" and retain the intrinsic semantics. Furthermore, semantic guidance improves the preservation of the original semantics by directing the reverse procedure. Ultimately, semantic fusion is designed to provide semantic guidance only during the concrete phase of the reverse process, preventing semantic ambiguity and artifacts while maintaining the guidance information. Extensive experiments validate the effectiveness and efficiency of our method in improving classical distillation techniques and significantly improving cross-architecture generalization, requiring processing time comparable to raw DiT on ImageNet ($256\times256$) with only 4.02 GB of GPU memory usage.

## 1 Introduction

Large-scale data is the fuel for deep learning, while deep learning is the engine that extracts its value, creating a virtuous cycle of mutual advancement Song et al. (2025). Nevertheless, the increasing complexity of data introduces significant risks of private information leakage Yu et al. (2023b) and imposes growing computational and storage burdens for deep learning models. *Dataset Distillation* Wang et al. (2018) (DD) mitigates these issues by compressing massive semantically rich datasets into compact, human-unreadable or non-raw (yet still readable) representations, while preserving essential information for learning. This process not only effectively safeguards sensitive information against leakage but also substantially reduces the overhead associated with model training.

However, classical DD framework employs bi-level optimization to generate distilled datasets. In the inner loop, the network is updated to evaluate classification performance, while the outer loop synthesizes images based on certain matching strategies. For instance, gradient Loo et al. (2023); Wang et al. (2023), distribution Deng et al. (2024); Zhang et al. (2024), and trajectory Du et al. (2023); Zhong et al. (2025) matching all perform the optimization process directly in the pixel space, which tends to excessively learn specific patterns that overfit on a prior architecture (such as ConvNet) employed during distillation Cazenavette et al. (2023). The resulting distilled images exhibit *abstract, noisy*, and *unrealistic* characteristics. Although such images are insightful and may enhance distillation performance, they suppress the expression of high-level visual semantics, thus falling into the cross-architecture generalization dilemma.

Some dual-time matching methods Yin et al. (2023); Shao et al. (2024) decouple the bi-level optimization into synthesis time and training time to achieve efficient processing of large-scale datasets. As shown in Fig. 4, although this makes the distilled image appear to have some semi-clear semantic expression, the entire optimization process still operates in pixel space, and the images remain

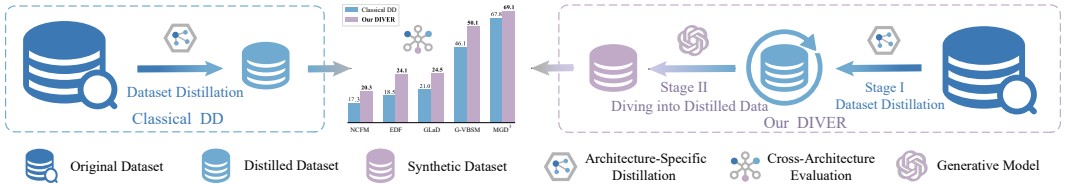

Figure 1: Comparison between classical single-stage DD and our proposed dual-stage DIVER. In stage I, DIVER is the same as classical DD. Mainly in stage II, we employ the pre-trained generative model to directly refine the distilled dataset, thereby synthesizing a new dataset termed synthetic dataset, significantly enhancing generalization capabilities of traditional techniques across various modes. The quantitative results are the average performance from Tab. 1 ~ Tab. 5, respectively.

unrealistic. Recent finding Sun et al. (2024) indicates that more realistic images facilitate cross-architecture generalization, and the performance of some existing DD methods is even worse than that of random selection Li et al. (2025). So the generative prior methods Cazenavette et al. (2023); Zhong et al. (2024) use GANs Karras et al. (2019; 2020) to synthesize images optimized in latent space, its effectiveness remains constrained by expensive inner loop matching mechanisms and inadequate semi-realistic semantic representation.

Consequently, several diffusion-based approaches Su et al. (2024); Gu et al. (2024); Chan-Santiago et al. (2025) leveraging the powerful synthesis capabilities of diffusion models have begun to emerge and demonstrated promising performance. However, these approaches represent a decisive departure from the well-established classical DD framework, even going so far as to fully discard its conventions in favor of strictly returning to the fundamentals of coreset selection. One might whimsically ask: *Is classical DD destined for the methodological museum, or does it retain untapped potential?*

As shown in Fig. 1 (left), previous paradigms typically adopt a single-stage DD (after extracting the distilled dataset from the original dataset, it is directly used for evaluation) mode. Although obtained images may exhibit limited natural fidelity, we believe that they contain rich semantic information necessary for model generalization, it's just that this meaningful information is suppressed by specific artifacts and unclear semantics, thus reducing generalization.

Inspired by this insight, as presented in Fig. 1 (right), we propose a novel dual-stage (after stage I distillation, the distilled dataset undergoes further refinement via an additional round of distillation and is subsequently evaluated) distillation framework, called DIVER. In stage I, DIVER is the same as classical DD. Mainly in stage II, we propose an innovative task *Diving into Distilled Data* (DDD), which employs the pre-trained diffusion model to progressively refine the distilled dataset via three semantic recovery strategies. As the resolution scheme for DDD, these strategies aim to recover expressive semantic masked and suppressed by prevalent specific patterns in the distillation output. Specifically, semantic inheritance filters out architecture-specific "noise" while distilling high-level semantic knowledge from distilled images into the latent space, ensuring that synthesized images maintain their core intrinsic semantics. Meanwhile, semantic guidance reinforces the preservation of the original semantics by steering the sampling process to produce *realistic*, *clear*, and *informative* outputs. Crucially, semantic fusion integrates conditional labels into inherited and guided latents to compensate for the lack of category information in the original features, thereby enhancing both sampling efficiency and quality. Generally, our contributions are summarized as follows:

- We propose a dual-stage distillation framework termed DIVER, decoupling the classic dataset distillation problem into DD and DDD.

- We formally introduce DDD. To the best of our knowledge, this is the first work to straightforwardly distill knowledge from distilled dataset into synthetic dataset without requiring access to the original dataset.

- We propose three recovery strategies integrated into pre-trained guided diffusion model, reviving the semantics suppressed by architecture-specific patterns in the distilled dataset without requiring additional training.

- Extensive experimental results demonstrate that DIVER effectively and efficiently enhances cross-architecture generalization as a plugin in traditional DD, with only 2.48s per image on a single RTX-4090 GPU using 4.02GB of memory.

## 2 PRELIMINARIES

### 2.1 CLASSICAL DATASET DISTILLATION

For a given large-scale original dataset $\mathcal{O} = \{(\boldsymbol{x}_i, \boldsymbol{y}_i)\}_{i=1}^{|\mathcal{O}|}$, dataset distillation aims to build a compact distilled dataset $\mathcal{D} = \{(\tilde{\boldsymbol{x}}_i, \tilde{\boldsymbol{y}}_i)\}_{i=1}^{|\mathcal{D}|}$ that extracts rich information from $\mathcal{O}$, such that models trained on $\mathcal{D}$ achieve performance within an acceptable deviation $\epsilon$ from those trained on $\mathcal{O}$, where $|\mathcal{D}| \ll |\mathcal{O}|$ and $\boldsymbol{x}_i, \tilde{\boldsymbol{x}}_i \sim P(\boldsymbol{x}), P(\tilde{\boldsymbol{x}})$ are the original and distilled images with the corresponding ground-truth labels $\boldsymbol{y}_i, \tilde{\boldsymbol{y}}_i \in \mathcal{Y} = \{1, 2, \cdots, C\}$, $P$ represents data distribution, $C$ is the number of categories. The capacity of $\mathcal{D}$ is determined by IPC (Images-Per-Class). This can be formulated as:

$$\sup_{\boldsymbol{x}, \tilde{\boldsymbol{x}} \sim P} |\mathcal{L}(\Phi_{\mathcal{O}}^p(\boldsymbol{x}), \boldsymbol{y}) - \mathcal{L}(\Phi_{\mathcal{D}}^p(\tilde{\boldsymbol{x}}), \tilde{\boldsymbol{y}})| \leqslant \epsilon, \tag{1}$$

where $\mathcal{L}$ denotes the loss function, $\Phi_{\mathcal{O}}^p$ and $\Phi_{\mathcal{D}}^p$ are two prior models with the same architecture but different initial parameters used during distillation.

To make this metric practically solvable, classical DD methods introduce $\Phi$ to obtain informative guidance from $\mathcal{O}$ and $\mathcal{D}$ in a chosen representation space, and iteratively optimize $\mathcal{D}$ accordingly:

$$\mathcal{D}^* = \arg\min_{\mathcal{D}} \mathcal{M}(\Phi_{\mathcal{O}}^p(\boldsymbol{x}), \Phi_{\mathcal{D}}^p(\tilde{\boldsymbol{x}})), \tag{2}$$

where $\mathcal{M}$ represents various matching metrics, such as distribution matching Loo et al. (2023), gradient matching Wang et al. (2023), and trajectory matching Du et al. (2023).

During distillation, $\Phi^p$ is required to be simple and efficient (e.g., ConvNet) because the optimization process is bi-level and performed directly in the pixel space, which is time-consuming and resource-intensive, and tends to excessively learn specific patterns that overfit on the prior architecture, while various complex architecture $\Phi^v$ for practical applications is preferred during evaluation.

### 2.2 GUIDED DIFFUSION MODEL

Diffusion models Ho et al. (2020); Song et al. (2020) are strong generative models that learn a mapping between Gaussian noise and the data distribution through the entire diffusion process including a forward noising process and a reverse denoising process. For the Latent Diffusion Model (LDM) Rombach et al. (2022), during training, the image $x_0$ is compressed from the pixel space $\mathcal{X}$ to the latent space $\mathcal{Z} : z_0 = \mathcal{E}(x_0)$ by using the VAE encoder $\mathcal{E}$. Then, the forward process gradually adds Gaussian noise $\epsilon \sim \mathcal{N}(0, I)$ to $z_0$: $z_t = \sqrt{\alpha_t} z_0 + \sqrt{1 - \alpha_t} \epsilon$. $\alpha_t$ controls the noise scale at step $t$.

During training, the denoising model learns to predict the noise $\epsilon_\theta(\hat{z}_t, t, c)$. During sampling, a Gaussian noise $\hat{z}_t$ is first initialized, the reverse process recovers the embedding $\hat{z}_0$ from $\epsilon_\theta$ at the end, where $c$ is a conditional input such as labels. Finally, the latent is decoded back to images: $\hat{x}_0 = \mathcal{F}(\hat{z}_0)$ by using the VAE decoder $\mathcal{F}$. For notation simplicity, the reverse process can be abstracted as: $\hat{z}_{t-1} = s(\hat{z}_t, \epsilon_\theta)$, where $s$ depends on the type of denoising. For example, in DDIM, $\hat{z}_{t-1}$ is sampled from distribution:

$$\hat{z}_{t-1} \sim \mathcal{N}(\sqrt{\alpha_{t-1}} \hat{z}_{0|t} + \sqrt{1 - \alpha_{t-1} - \sigma_t^2} \epsilon_\theta(\hat{z}_t, t, c), \sigma_t^2 I) \tag{3}$$

where $\sigma_t$ is predefined noise factor, $\hat{z}_{0|t}$ is the clean latent predicted based on $\hat{z}_t$:

$$\hat{z}_{0|t} = \frac{1}{\sqrt{\alpha_t}} (\hat{z}_t - \sqrt{1 - \alpha_t} \epsilon_\theta(\hat{z}_t, t, c)) \tag{4}$$

Traditional diffusion models commonly utilize conditioning mechanisms Ho & Salimans (2022); Wang et al. (2022b) to generate customized outputs based on targeted user inputs, including textual prompts or categorical labels. Although these approaches work well under various constraints, they remain computationally expensive due to the need for model retraining. Recent works Lin et al. (2025); Bansal et al. (2023); Yu et al. (2023a) have demonstrated the effectiveness of employing frozen pre-trained diffusion models as base architectures, where the sampling process is adaptively guided by learned feedback mechanisms to produce target-specific outputs according to user requirements. The reverse process in guided diffusion can typically be implemented as follows:

$$\hat{z}_{t-1} = s(\hat{z}_t, t, \epsilon_\theta) - \gamma * \nabla_{\hat{z}_t} \mathcal{G}_t(\hat{z}_t) \tag{5}$$

where $\mathcal{G}$ is the guidance function used to induce the generation of customized samples, and $\gamma$ is a guidance factor modulating the guidance intensity.

## 3 METHOD

The comprehensive pipeline of our proposed DIVER is illustrated in Fig. 6 and Algorithm 1 in the Appendix. In Section 3.1, we decouple the classic dataset distillation problem into DD and DDD. In Section 3.2, we use traditional DD to extract abundant information from a cumbersome original dataset into a tiny distilled dataset. In Section 3.3, we propose DDD integrating three semantic recovery strategies, which leverages the pre-trained VAE and diffusion models to filter out architecture-specific patterns in the distilled dataset and injects category information, efficiently release high-level expressive semantics to improve generalization performance.

### 3.1 DECOUPLED DATASET DISTILLATION

An effective distilled dataset should demonstrate strong performance across various network architectures $\Phi^v$ rather than being tailored to a specific one $\Phi^p$. In conventional DD approaches, the distilled dataset $\mathcal{D}^*$ extracted in accordance with the objective of Eqn. 2 learns specific patterns on a prior architecture $\Phi^p$, thereby inhibiting expressive semantics that favor generalization. This outcome is attributed to the different proxy models $\Phi$ employed during distillation and evaluation, which prevented the optimization process from reaching the global optimum.

In $D^*$, overfitting the specific architecture causes the distilled images to exhibit *abstract*, *noisy*, and *unrealistic* properties. They inadvertently suppress the expression of semantics, leading to a cross-architecture generalization bottleneck, which motivates us to perform in-depth exploration to mitigate the impact. As illustrated in Eqn. 6, we decouple the classic dataset distillation problem into DD (Dataset Distillation) and DDD (Diving into Distilled Data).

***Define DDD.*** Given a distilled dataset $\mathcal{D}^*$ compressed from the original dataset $\mathcal{O}$, the goal of DDD is to refine $\mathcal{D}^*$ into a synthetic dataset $\mathcal{S}$ to mitigating the impact of the specific architecture $\Phi^p$ on generalization and minimize the objective on various architectures $\Phi^v$:

$$\mathcal{S}^* = \arg\min_{\mathcal{S}} \mathcal{M}(\Phi^v_{\mathcal{O}}(\boldsymbol{x}), \Phi^v_{\mathcal{S}}(\mathcal{H}_{\mathcal{D}_*}(\tilde{\boldsymbol{x}}))) \qquad s.t. \ \mathcal{D}^* \leftarrow \mathcal{O}, \tag{6}$$

where $\mathcal{H}_{\mathcal{D}_*}(\tilde{\boldsymbol{x}})$ is the synthetic image, the size $|\mathcal{S}| = |\mathcal{D}| \ll |\mathcal{O}|$. $\mathcal{H}$ denotes the semantic recovery strategies, which is designed to refine the semantics of distilled images, unlocking their suppressed potential for generalization. Generally speaking, while the initial objective of Eqn. 2 is designed for all $\Phi$, its practical effectiveness is primarily observed at $\Phi^p$. Our goal is to refine the distilled images to ensure its applicability at $\Phi^v$ as well.

### 3.2 STAGE I: DATASET DISTILLATION

Stage I comprises the entire process of classical DD aiming to obtain the distilled dataset, applicable to most existing distillation techniques. This stage is not essential in our framework, as: (1) Our core objective is to obtain the distilled dataset rather than intermediate results from the distillation process. (2) Due to privacy reasons, the original dataset is not released by the institution. Therefore, when the distilled dataset already exists, we directly proceed to stage II without executing stage I.

### 3.3 STAGE II: DIVING INTO DISTILLED DATA

#### 3.3.1 SEMANTIC INHERITANCE

Previous studies Cazenavette et al. (2023) show that images distilled by traditional methods have low-level patterns that overfitting to a specific architecture. And the canonical hierarchical feature extraction principle in neural networks shows shallow layers predominantly encode low-level patterns (e.g., textures, edges and noise) and deeper layers capture high-level semantics Zeiler & Fergus (2014). We draw inspiration from LDM, and propose the semantic inheritance strategy that projects the distilled image $x_0$ into deep latent code $z_0$ via a pre-trained image encoder. Through this process, $z_0$ inherently inherits the high-level semantic representations from the distilled image while simultaneously filtering out architecture-specific "noise". Then, we add noise $t_f$ steps to $z_0$:

$$z_{t_f} = \sqrt{\alpha_{t_f}} z_0 + \sqrt{1 - \alpha_{t_f}} \epsilon \tag{7}$$

We initialize the latent code of sampling process with $\hat{z}_{t_r} = z_{t_f}$, deliberately replacing conventional random noise initialization to preserve structural semantics throughout the diffusion process. $t_r$ is the number of denoising steps. This operation additionally serves as a regularization mechanism, constraining the sampling trajectory to remain proximate to the initial embedding space of distilled images, thereby mitigating potential semantic drift during generation.

The choice of $t_f$ is critical, if it is too large, $z_0$ will completely obey the Gaussian distribution, causing the semantics to disappear, and if it is too small, $z_0$ will deviate from the distribution, which is contrary to the assumption of diffusion and will lead to poor sampling quality Zhao et al. (2025).

### 3.3.2 SEMANTIC GUIDANCE

During the image synthesis phase, our goal is to generate the high-quality image that satisfies the semantics of the specific distilled image. However, due to the continuous injection of conditional label information during the reverse process, latent code using semantic inheritance inevitably suffers from information degradation. To compensate, we introduce semantic guidance that actively reinforces semantic retention from the distilled image in the synthetic image. We design the guidance function according to Eqn. 5.

$$\mathcal{G}_t = (\hat{z}_t - z_0)^2 \sigma_t / 2 \tag{8}$$

The purpose of $\mathcal{G}_t$ is to maintain the semantics inherited from the distilled dataset so that the fusion of label semantics does not deviate too much from the initial semantics.

### 3.3.3 SEMANTIC FUSION

Recent studies Yu et al. (2023a); Chen et al. (2025) have demonstrated that the sampling process of diffusion models can be divided into three distinct phases: Chaotic Phase (CP, $t_r \sim t_h$), Semantic Phase (SP, $t_h \sim t_l$), and Refinement Phase (RP, $t_l \sim 1$), with the majority of semantic content being formed during SP. Inspired by this observation, we propose to merge SG and conditional labels based on inher-

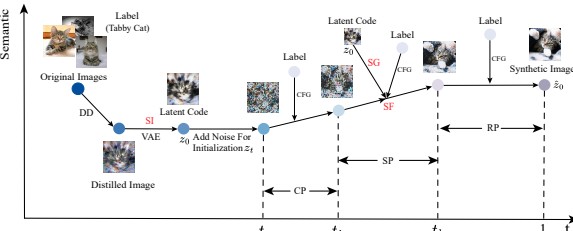

Figure 2: Semantic evolution of the entire process.

ited latents exclusively during this critical SP. Our experiments find that this targeted approach not only enhances sampling efficiency but also improves quality.

Fig. 2 illustrates the entire semantic evolution process of our DIVER. Initially, SI inherits the valuable information from the distilled dataset. It then adds noise for $t_f$ steps according to Eqn. 7 to initialize the latent, which preserves information while satisfying the standard diffusion mapping from a Gaussian distribution to the data distribution. Inherited semantics persist throughout the entire phase. Subsequently, Classifier-Free Guidance (CFG) is applied throughout the reverse process. Specifically, we use SG only in the SP, which involves fusing inherited semantics, conditional labels, and guidance, while using CFG only in the CP and RP. This design aims to produce clear semantics while preserving information from the distilled dataset, thus preventing semantic ambiguity and artifacts as shown in Fig. 4 (right).

## 4 EXPERIMENTS

### 4.1 EXPERIMENTAL SETUP

**Datasets.** Experiments are conducted on large-scale and high-resolution datasets. We evaluate the performance on the complete ImageNet-1K dataset Deng et al. (2009) with 224×224 resolution and its twelve subsets (e.g., ImageNette, ImageA and ImageIDC) Howard (2019) with 128×128 resolution. All images are resized to 256×256 in our method.

**Baselines and evaluation.** We compare DIVER with several state-of-the-art DD methods including distribution matching {DM Zhao & Bilen (2023), NCFM Wang et al. (2025b)}, gradient or trajectory matching {MTT Cazenavette et al. (2022), EDF Wang et al. (2025a), DC Zhao et al. (2020)},

Table 1: ImageNet Subsets (ImageFruit ∼ ImageYellow, 128x128) performance on unseen architectures. The results are averaged from 5 times on the real validation sets.

| Method | IPC = 1 | | | | | | IPC = 10 | | | | | |
|---|---|---|---|---|---|---|---|---|---|---|---|---|
| | Fruit | Woof | Meow | Squawk | Nette | Yellow | Fruit | Woof | Meow | Squawk | Nette | Yellow |
| DM | 11.3±1.4 | 10.7±1.1 | 11.7±1.5 | 12.7±1.4 | 13.3±2.0 | 12.5±1.5 | 19.3±1.5 | 14.5±0.8 | 17.6±1.7 | 24.4±2.3 | 23.7±1.1 | 26.4±1.5 |
| **Ours** | **18.5**±1.9 | **13.1**±0.8 | **14.4**±1.7 | **18.0**±1.6 | **21.6**±2.2 | **22.4**±1.8 | **22.9**±1.9 | **16.8**±1.6 | **20.0**±1.6 | **28.1**±3.0 | **28.6**±2.9 | **29.0**±2.1 |
| NCFM | 17.1±1.6 | 12.0±1.6 | 12.3±1.5 | 16.8±1.9 | 16.1±1.5 | 17.3±2.6 | 20.5±2.5 | 14.3±1.7 | 15.7±2.3 | 22.6±2.8 | 21.0±2.6 | 22.5±2.8 |
| **Ours** | **18.8**±1.7 | **12.5**±1.3 | **15.8**±1.5 | **18.0**±1.2 | **18.1**±2.0 | **21.0**±2.0 | **25.5**±1.6 | **16.2**±1.4 | **20.8**±1.7 | **24.0**±2.1 | **25.3**±2.0 | **28.0**±2.6 |
| MTT | 15.4±1.6 | 13.8±1.4 | 14.1±1.7 | 14.0±1.5 | 17.7±1.9 | 17.3±1.7 | 18.9±1.4 | 15.9±1.5 | 16.1±1.4 | 21.7±2.5 | 20.7±1.8 | 19.1±0.9 |
| **Ours** | **22.3**±1.8 | **16.2**±1.1 | **15.7**±1.8 | **17.2**±1.6 | **20.3**±1.5 | **20.2**±1.6 | **29.8**±2.0 | **21.5**±1.6 | **26.7**±1.7 | **33.8**±1.4 | **34.3**±1.4 | **34.8**±1.6 |
| EDF | 16.2±1.8 | 15.2±1.7 | 16.2±1.6 | 16.5±1.9 | 18.0±1.5 | 18.8±2.6 | 20.5±1.5 | 17.2±1.6 | 17.5±1.8 | 23.2±2.1 | 21.2±1.8 | 21.1±1.5 |
| **Ours** | **20.3**±1.9 | **18.4**±1.2 | **17.5**±2.1 | **19.4**±1.6 | **22.3**±2.1 | **21.8**±1.8 | **26.3**±2.0 | **23.8**±1.8 | **25.2**±1.6 | **34.5**±2.3 | **28.5**±1.4 | **31.6**±1.9 |

Table 2: ImageNet Subsets (ImageA ∼ ImageE, 128×128) performance on unseen architectures across different distillation modes under IPC=1.

| Alg. | Mode | A | B | C | D | E |
|---|---|---|---|---|---|---|
| DM | DD | 24.0±2.7 | 16.7±2.5 | 18.1±2.4 | 14.5±1.7 | 15.4±1.5 |
| | GLaD | 25.3±2.1 | 20.1±1.8 | 19.3±1.6 | 18.3±1.8 | 14.8±2.6 |
| | **Ours** | **28.6**±2.0 | **26.1**±1.9 | **21.5**±1.8 | **21.1**±2.4 | **16.7**±2.2 |
| DC | DD | 24.9±3.1 | 21.3±2.4 | 21.2±1.5 | 16.9±1.6 | 17.1±2.0 |
| | GLaD | 27.4±2.1 | 23.7±1.7 | 22.5±2.2 | 17.1±1.1 | 18.6±2.8 |
| | **Ours** | **30.5**±2.2 | **27.8**±2.1 | **25.1**±1.8 | **21.4**±2.0 | **20.6**±1.6 |
| MTT | DD | 25.4±2.4 | 21.0±1.5 | 19.4±1.7 | 16.1±2.5 | 16.3±2.8 |
| | GLaD | 29.1±1.5 | 22.5±3.2 | 19.1±2.9 | 20.0±1.4 | 17.1±1.9 |
| | **Ours** | **31.6**±2.7 | **28.7**±1.8 | **24.9**±2.3 | **22.1**±2.2 | **20.2**±2.1 |

Table 3: Dual-time matching methods with our DIVER on ImageNet-1K (224×224). We cite the experimental results from G-VBSM.

| Alg. | IPC | Mode | RN18 | RN50 | RN101 |
|---|---|---|---|---|---|
| SRe$^2$L | 10 | DD | 21.3±0.6 | 28.4±0.1 | 30.9±0.1 |
| | | **Ours** | **24.5**±0.4 | **31.2**±0.2 | **32.4**±0.3 |
| | 50 | DD | 46.8±0.2 | 55.6±0.3 | 60.8±0.5 |
| | | **Ours** | **54.0**±0.5 | **61.1**±0.2 | **61.3**±0.4 |
| G-VBSM | 10 | DD | 31.4±0.5 | 35.4±0.8 | 38.2±0.4 |
| | | **Ours** | **35.1**±0.4 | **40.4**±0.5 | **40.1**±0.4 |
| | 50 | DD | 51.8±0.4 | 58.7±0.3 | 61.0±0.4 |
| | | **Ours** | **57.2**±0.6 | **64.2**±0.4 | **63.7**±0.3 |

generative prior {GLaD Cazenavette et al. (2023)}, dual-time matching {SRe$^2$L Yin et al. (2023), G-VBSM Shao et al. (2024)}, and diffusion-based {Minimax Gu et al. (2024) , D$^4$M Su et al. (2024), MGD$^3$ Chan-Santiago et al. (2025)} methods under the same evaluation configuration. We directly use the public distilled datasets of MTT, SRe$^2$L, and G-VBSM, and the others use the official code to obtain the distilled datasets (NCFM w/o sampling network and using "Mix" initialization). We evaluate the methods using hard-label protocol from prior studies Zhao & Bilen (2023); Sajedi et al. (2023); Cazenavette et al. (2023) excluding the methods used on ImageNet-1K, which use soft-label protocol with KL divergence loss. The reported results represent averages of 5 trials on the subsets and 3 trials on the full ImageNet-1K dataset.

**Implementation details.** We utilize the pre-trained Diffusion Transformer (DiT-XL/2, the image size is 256×256) and VAE model (vae-ft-mse) introduced by Peebles & Xie (2023), originally trained on ImageNet-1K. We use the conditioning labels and 50 sampling steps with classifier-free guidance. For forward process, we set $t_f$ to 25. For semantic fusion, we set $t_h$ and $t_l$ to 40 and 25. The scaling factor $\gamma$ is set to 0.1 in all methods except NCFM which is set to 0.02. All the experimental results of our method can be obtained on a single NVIDIA 4090 or A800 GPU.

## 4.2 COMPARISON WITH STATE-OF-THE-ART METHODS

**Cross-Architecture Generalization.** A critical limitation of existing DD techniques lies in their inadequate cross-architecture generalization capability, which serves as a key indicator of whether the method genuinely captures the underlying classification task semantics rather than merely overfitting to specific architectural features. As presented in Tab. 1, we show generalization results for distribution-based and trajectory-based methods with and without our DIVER. All distilled datasets generated by classic DD are distilled using a specific ConvNet. We train ResNet18, ShuffleNet-V2, MobileNet-V2, EfficientNet-B0, and ViT-b/16 Dosovitskiy et al. (2020) on distilled dataset from scratch and evaluate cross-architecture generalization. The experimental results are averaged from 5 times. Across all tested datasets, the incorporation of semantic recovery in DIVER consistently improve the performance of the generalization of all methods, with improvements ranging from marginal to substantial.

Table 4: Performance comparison over ResNet-18 with state-of-the-art DD on ImageNet-1K. Our method can be effectively integrated into both the original DiT-based and Minimax-tuned MGD$^3$.

| IPC | SRe$^2$L | G-VBSM | RDED | DiT | Minimax | D$^4$M | MGD$^3$ | MGD$^3$+Ours | MGD$^3$+Minimax+Ours |
|---|---|---|---|---|---|---|---|---|---|
| 10 | 21.3$_{\pm 0.6}$ | 31.4$_{\pm 0.5}$ | 42.0$_{\pm 0.1}$ | 39.6$_{\pm 0.4}$ | 44.3$_{\pm 0.5}$ | 27.9$_{\pm 0.2}$ | 45.8$_{\pm 0.3}$ | 46.4$_{\pm 0.3}$ | **46.9$_{\pm 0.5}$** |
| 50 | 46.8$_{\pm 0.2}$ | 51.8$_{\pm 0.4}$ | 56.5$_{\pm 0.1}$ | 52.9$_{\pm 0.6}$ | 58.6$_{\pm 0.3}$ | 55.2$_{\pm 0.1}$ | 60.2$_{\pm 0.1}$ | 60.6$_{\pm 0.4}$ | **61.0$_{\pm 0.2}$** |

Table 5: Our semantic recovery strategies are applied to the diffusion-based prototype learning. We cite the experimental results from MGD$^3$.

| Dataset | ImageNette | | | ImageIDC | | |
|---|---|---|---|---|---|---|
| IPC | 10 | 20 | 50 | 10 | 20 | 50 |
| D$^4$M | 59.1 | 64.3 | 70.2 | 52.3 | 55.5 | 62.7 |
| D$^4$M+Ours | 62.5 | 65.8 | 75.1 | 53.7 | 58.9 | 67.4 |
| MGD$^3$ | 66.4 | 71.2 | 79.5 | 55.9 | 61.9 | 72.1 |
| MGD$^3$+Ours | **67.2** | **72.4** | **81.2** | **56.6** | **63.0** | **73.9** |

Table 6: Comparison results of optimized DD (NCFM) with diverse initializations versus our DIVER on ImageSquawk under IPC=1.

| Init. | Mode | RN18 | SN-V2 | EN-B0 | ViT |
|---|---|---|---|---|---|
| Random | DD | 21.2$_{\pm 3.3}$ | 18.0$_{\pm 3.3}$ | 14.1$_{\pm 2.1}$ | 17.2$_{\pm 0.6}$ |
| | Ours | **23.1$_{\pm 2.5}$** | **19.6$_{\pm 1.8}$** | **18.7$_{\pm 1.9}$** | **23.6$_{\pm 0.4}$** |
| Noise | DD | 14.4$_{\pm 1.9}$ | 15.6$_{\pm 2.2}$ | 11.0$_{\pm 1.8}$ | 19.4$_{\pm 1.0}$ |
| | Ours | **20.8$_{\pm 2.6}$** | **21.2$_{\pm 3.4}$** | **15.5$_{\pm 0.9}$** | **21.7$_{\pm 1.0}$** |
| Mix | DD | 17.6$_{\pm 1.5}$ | 15.3$_{\pm 2.0}$ | 13.6$_{\pm 0.2}$ | 18.9$_{\pm 1.8}$ |
| | Ours | **22.6$_{\pm 0.8}$** | **18.5$_{\pm 2.0}$** | **14.2$_{\pm 2.2}$** | **19.8$_{\pm 1.1}$** |

**Generative Prior.** Although GLaD mitigates the architecture-specific pattern by training latent codes instead of pixel space through the traditional DD paradigm with a generator, it remains reliant on specific ConvNet architectures and requires computing and storing generator gradients during training. This substantial overhead limits its scalability to high-IPC. In contrast, our method primarily builds upon the distilled dataset, requiring neither gradient computation nor access to the original dataset. Such efficiency is remarkable, and as demonstrated in Tab. 2, our approach also achieves significantly superior generalization compared to GLaD.

**Dual-Time Matching.** SRe$^2$L and G-VBSM use squeeze-recover-relabel process to decouple the bi-level optimization into synthesis time and training time, which scales to high-resolution datasets with low training cost and memory consumption. We integrate their distilled datasets into our framework to synthesize new datasets and evaluate their performance in Tab. 3. We employ ResNet18 as the recovery model and use soft labels (KD) to train and evaluate the ResNet-{18, 50, 101}, respectively. Our approach consistently outperforms the original method across multiple squeeze models. Notably, as the squeeze model becomes stronger (RN18 → RN101), our performance advantage gradually diminishes. This suggests a transition in the dominant factor of performance gains from the information content of the compressed data to the capability of the squeeze model trained on the original dataset. When RN101 serves as the squeeze model, the performance gap becomes marginal.

**Diffusion-Based Method.** D$^4$M and MGD$^3$ encode the full dataset into the latent space via a VAE for prototype learning, and expect the diffusion model to generate representative prototypes as the synthetic dataset. However, their ability to preserve prototype information remains limited. We attempt to apply their synthetic datasets, which contain relatively clear semantic structures, within our framework, but this results in performance degradation. This is mainly attributed to (1) the encoding of VAE from high-dimensional to low-dimensional space itself loses information, and (2) uncertainty introduced by diffusion. The detailed results and analysis are provided in Tab. 11 of the appendix. Therefore, we follow their paradigm by encoding the full dataset into the latent space for prototype learning, and subsequently apply our SG and SF modules to generate the synthetic dataset. As shown in Tab. 4 and Tab. 5, our method can stably preserve the prototype information learned from the original datasets on ImageNet-1K and its subsets. Furthermore, after fine-tuning with Minimax, we achieve further performance improvement, demonstrating the superior scalability.

## 4.3 ABLATION STUDIES

**Effect of Each Module.** As illustrated in Tab. 7, we incrementally evaluate the impact of each proposed semantic recovery component under different IPC settings. From a holistic perspective, every individual component plays a critical role in determining and enhancing the system's overall performance. From a local perspective, the cross-architecture performance of classical DD is merely comparable to random selection (14.1% vs. 15.4%), and even inferior to it under high IPC conditions (18.9% < 19.6%). Both SI and SG individually demonstrate significant performance improvements, while their combination lead to a saturation effect, resulting in only marginal additional

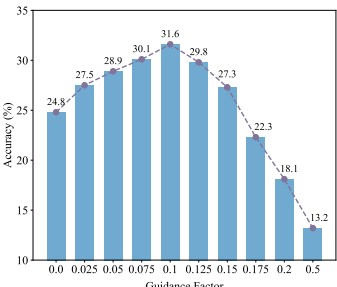 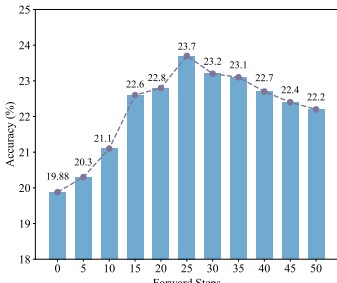 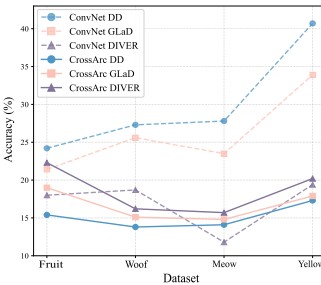

Figure 3: (**Left**) The effect of guidance factor on performance on ImageYellow (IPC=10) with EDF. (**Medium**) The effect of applying different forward steps to the inherited latent on performance on ImageFruit (IPC=10) with NCFM (only SI). (**Right**) Performance of DD (MTT), GLaD and our DIVER under IPC 1 on the specific ConvNet and across heterogeneous architectures.

gains. Notably, the raw DiT without any recovery modules also achieves moderate performance enhancement, which can be attributed to the fact that diffusion models enhance generalization by generating samples that closely follow the original data distribution while exhibiting greater diversity, thereby covering underrepresented regions. Additionally, the inherent smoothness of their generative process yields "cleaner" samples (e.g., denoised data), which implicitly regularizes training and mitigates overfitting Song & Ermon (2019); Ho et al. (2020).

In addition, while incorporating randomly selected images into our framework also yields certain improvements, it does not surpass DiT. This indirectly confirms that our approach does not rely on DiT, but is closely tied to the semantic information provided by the images. In fact, the distillation dataset implicitly encapsulates the semantic knowledge of the entire dataset. Surprisingly, using SF with fused semantics only in the partial phase to reduce guidance yields considerable gains. We will discuss this reason in detail in Section 4.4.

**Guidance Factor.** As presented in Fig. 3 (left), when the guidance factor is too small, semantic guidance becomes insufficient. Conversely, when the guidance factor is excessively high, the sampling process over-relies on the latent

Table 7: The isolated and combinatorial effects of constituent elements in our proposed semantic recovery. Random* indicates randomly selected images that are incorporated into our framework. The results are on the ImageFruit with IPC=1 and IPC=10. The Alg. is MTT.

| Mode | SI | SG | SF | IPC=1 | IPC=10 |
|---|---|---|---|---|---|
| Random | - | - | - | $14.1_{\pm1.4}$ | $19.6_{\pm1.8}$ |
| DD | - | - | - | $15.4_{\pm1.6}$ | $18.9_{\pm1.4}$ |
| Random* | ✓ | ✓ | ✓ | $14.6_{\pm1.8}$ | $21.7_{\pm1.6}$ |
| **Ours** | × | × | × | $17.8_{\pm1.2}$ | $23.4_{\pm1.3}$ |
| | ✓ | × | × | $19.5_{\pm1.4}$ | $26.2_{\pm1.5}$ |
| | × | ✓ | × | $20.4_{\pm1.7}$ | $27.8_{\pm1.9}$ |
| | ✓ | ✓ | × | $21.1_{\pm1.9}$ | $28.3_{\pm1.6}$ |
| | ✓ | ✓ | ✓ | $\mathbf{22.3_{\pm1.8}}$ | $\mathbf{29.8_{\pm2.0}}$ |

codes inherited from the distilled dataset, thereby diminishing the conditioning effect of the label guidance. This scenario is particularly catastrophic for synthetic dataset, leading to a significant degradation in performance.

**Forward Steps.** As presented in Fig. 3 (medium), we apply different forward steps to the inherited latent to analyze its impact. As the number of noise-adding steps increases, performance begins to improve, reaching its optimal level at 25 steps. However, further increasing the steps results in a decline in performance, as the latent variables gradually approach a pure Gaussian distribution, causing the loss of original features. Overall, an intermediate number of steps achieve a balance between initial Gaussian distribution alignment and feature preservation, leading to superior performance.

**Initialization of Distilled Images.** To assess the robustness of DIVER, we obtain distilled datasets optimized under different initial image settings, including: (1) random selection from original images, (2) Gaussian noise, and (3) mixed images composited from four source images, and all integrated into our framework. As shown in Tab. 6, our method consistently improves the generalization of distilled datasets optimized under various conditions, demonstrating its superiority and stability.

**Robustness across Different Diffusion Models.** As evaluated in Tab. 8, our method demonstrates consistent performance gains across different diffusion models including Stable Diffusion V-1.5(SD-V1.5), DiT and SiT, highlighting the robustness of DIVER. The smaller gain with SD-V1.5 is at-

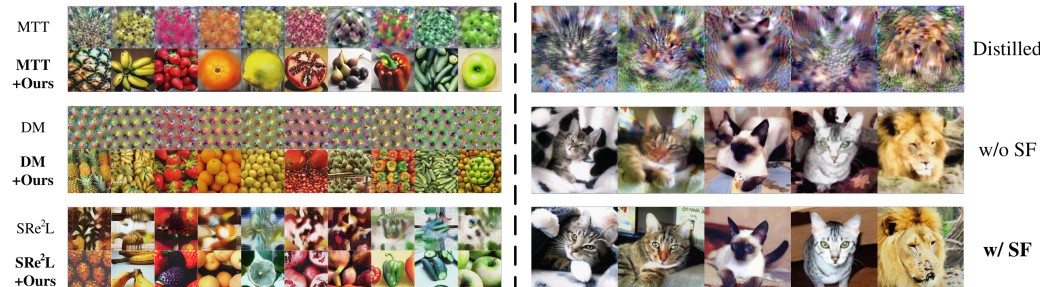

Figure 4: (**Left**) Comparison of synthetic images from different methods with and without our DIVER on ImageFruit. Our approach recovers expressive semantics and is more realistic. (**Right**) Comparison of images generated with (semantic-phase fusion) and without (full-phase fusion) our SF on ImageMeow. SF enhances category clarity and fidelity.

Table 8: Sensitivity of our DIVER to different diffusion models. The results are on the ImageNette with MTT.

| IPC | DD | Ours | | |
|---|---|---|---|---|
| | | SD-V1.5 | DiT | SiT |
| 1 | $17.7_{\pm1.9}$ | $19.1_{\pm1.3}$ | $20.3_{\pm1.5}$ | $20.2_{\pm1.7}$ |
| 10 | $20.7_{\pm1.8}$ | $26.5_{\pm2.1}$ | $34.3_{\pm1.4}$ | $33.1_{\pm2.3}$ |

Table 9: Performance of reconstructing distilled images using only VAE. The results are on the ImageMeow with IPC=10.

| Method | Architecture | Distilled | Reconstructed | Ours |
|---|---|---|---|---|
| DM | ConvNet | $23.2_{\pm0.8}$ | $21.2_{\pm1.2}$ | $19.6_{\pm0.9}$ |
| | CrossArc | $17.6_{\pm1.7}$ | $18.9_{\pm1.0}$ | $20.0_{\pm1.6}$ |
| MTT | ConvNet | $37.1_{\pm1.3}$ | $31.6_{\pm0.7}$ | $28.3_{\pm1.1}$ |
| | CrossArc | $16.1_{\pm1.4}$ | $21.8_{\pm1.9}$ | $26.7_{\pm1.7}$ |

tributed to architectural discrepancies (U-Net vs. Transformer) and its different pre-training dataset (not ImageNe-1K) in contrast to the more substantial improvements observed with DiT and SiT.

**Role of VAE.** As shown in Tab. 9, the reconstructed images also demonstrate promising generalization capability, exhibiting only a slight performance drop on the prior ConvNet. In contrast, our synthetic images achieve further improvement in cross-architecture generalization, while their distillation performance on ConvNet decreases significantly, reflecting a structured trade-off. Together with the visualization results in Fig. 7 of the appendix, we observe that the architecture-specific patterns consist of **noise coverage** and **semantic degradation**. The VAE primarily filters out "noise", whereas the diffusion model recoveries semantic information, and their joint effect enhances cross-architecture generalization performance.

## 4.4 ANALYSIS

**Performance on ConvNet.** In Fig. 3 (right), we show the performance of the images on the backbone ConvNet for DD. DIVER shows a measurable performance decline while maintaining competitive accuracy. We consider this an acceptable trade-off given its superior generalization. This performance decline primarily stems from two factors: (1) The encoder maps the distilled images into a deep latent space, filtering out architecture-specific patterns (ConvNet) while preserving informative semantics. This is evidenced by our experimental finding in Tab. 9 that directly decoding these latent representations (without diffusion model) still yields considerable generalization gains. (2) The diffusion model further eliminates residual architecture-specific information (degraded semantics) in the latent space through denoising steps, thereby injecting clear semantic information to enhance generalization. Glad shows only a slight decrease in performance on ConvNet, mainly due to its one-stage architecture, which also utilizes information provided by ConvNet during distillation. However, its generalization is mediocre, and the introduction of GAN significantly increases training overhead, limiting its application. Currently, it is primarily focused on 1 IPC settings.

**Visualization.** As shown in Fig. 4 (left), we compare image generation results from classical DD and their integration into our DIVER on ImageFruit. DIVER distills semantic information from abstract distilled images rich in specific patterns, yielding more realistic and informative images with comprehensive semantic coverage and similar semantic representation.

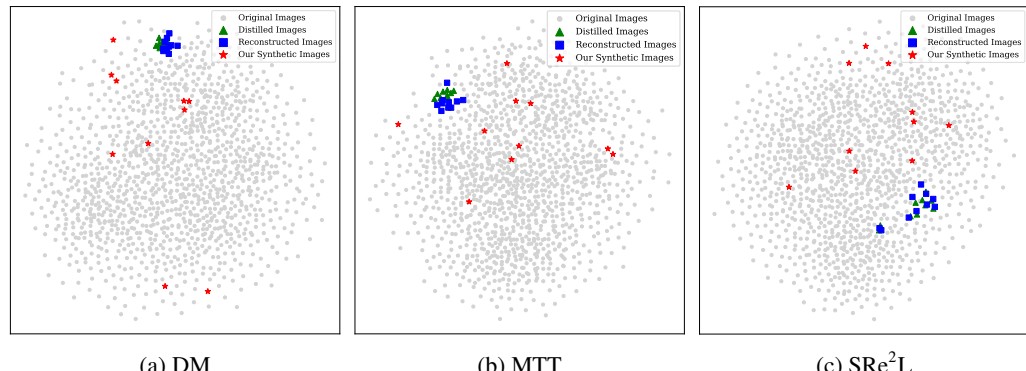

(a) DM          (b) MTT          (c) SRe$^2$L

Figure 5: t-SNE visualization of original images, distilled images, reconstructed images (obtained through direct VAE encoding and decoding without the diffusion model), and our synthetic images. The results are presented for the first class (Tench) of ImageNet under 10 IPC.

As shown in Fig. 4 (right), we compare synthetic images generated with (semantic-phase fusion) and without (full-phase fusion) our proposed SF on ImageMeow. SF makes the category information clearer and has better fidelity, while its absence leads to blurring and artifacts in synthetic images, which also accounts for the performance gap described in Tab. 7. This phenomenon primarily stems from two factors: (1) The latent code of the distilled dataset still retains residual architecture-specific patterns, where overfitting exacerbates image distortion and artifacts, inevitably degrading performance. (2) Full-phase guidance obstructs the injection of conditional category information and also runs the risk of overfitting, ultimately compromising visual fidelity. Similar to the images we synthesized on ImageFruit in Fig. 4 (left), our synthetic images exhibit semantic representations similar to those in the initial distilled dataset. This semantic coverage extends across nearly the entire pixel space, appearing as though previously suppressed semantics have been fully liberated, resulting in outputs that seem authentic and informative. This is the same as our initial goal, which will greatly improve cross-architecture generalization ability.

The t-SNE Fig. 5 shows that both distilled and reconstructed images are locally clustered and compact, whereas our synthetic images exhibit greater diversity within the original distribution, which is beneficial for generalization.

**Computational and Memory Costs.** We leverage pre-trained diffusion model to refine the distilled dataset, achieving superior generalization performance without incurring additional computational costs for fine-tuning. Notably, the architecture-free nature of DIVER guarantees fixed time and GPU memory requirements during synthesis. For our method, SI only needs to encode distilled images into latent codes, incurring negligible computational overhead. SG employs Eqn. 8 to compute gradients directly in the latent space for sampling guidance, with this additional operation introducing only minimal computational burden. To further minimize the computational cost, we set $\nabla_{\hat{z}_t}\mathcal{G}_t(\hat{z}_t) = (\hat{z}_t - z_0)\sigma_t$, eliminating gradient calculations entirely during sampling. Combined with SF, our approach achieves sampling time comparable to raw DiT (2.41s per image). Our method takes only 2.48s per image processing on ImageNet (256×256) with 4.02 GB memory usage on a single RTX-4090 GPU, demonstrating the computational and memory efficiency of our framework.

## 5 CONCLUSION

In this paper, we propose the novel task of "Diving into Distilled Data" for the first time, where we enhance remarkable cross-architecture generalization ability by recovering the expressive semantics suppressed by specific patterns in the distilled dataset. Our method can be used as a plug-in to directly optimize the distilled dataset generated by classical dataset distillation in a raw data-free and training-free manner. This efficient dual-stage paradigm achieves high model performance with minimal storage and computational overhead, which holds significant implications for resource-constrained deep learning in edge computing environments.

## 6 ETHICS STATEMENT

This work adheres to the ICLR Code of Ethics. Ethical approval was not required as the research did not involve human participants or animal studies. The data used were publicly available under terms that permit academic research, and contained no personal identifiers. We proactively addressed potential biases in our methodology. The study design raises no privacy or security issues, upholding our commitment to responsible research practices.

## 7 REPRODUCIBILITY STATEMENT

Our work is reproducible. The information necessary to reproduce the results of this paper is contained within the manuscript, and the experimental setup (including hyperparameters and evaluation metrics) is fully specified in Section 4.1. The source code and script files for implementation and experiments are provided in the supplementary materials. Since our method is available as a plug-in, it is easy to reproduce our results using existing publicly available distilled datasets.

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

# A APPENDIX

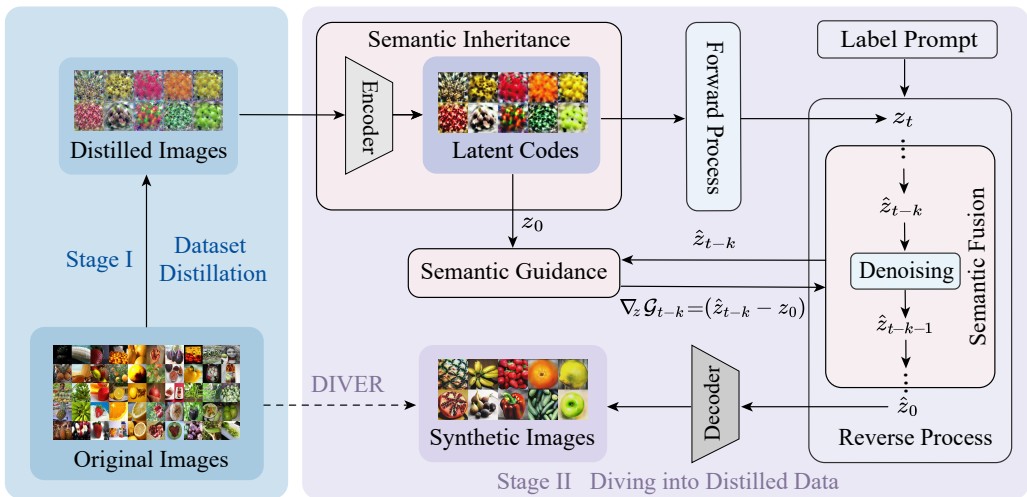

Figure 6: An overview of DIVER. Semantic inheritance filters out architecture-specific "noise" and distills high-level semantic of distilled images into the latent space, retaining the initial semantics. Semantic guidance enhances the preservation of original semantics by directing the sampling procedure to generate realistic and informative images. Semantic fusion fuses conditional labels with inherited and guided semantics only during specific stages of the reverse process, thereby enhancing both sampling efficiency and quality.

---

**Algorithm 1:** DIVER: Diving Deeper into Distilled Data via Expressive Semantic Recovery

---

**Input:** Pre-trained diffusion model $\epsilon_\theta$, VAE encoder $\mathcal{E}$ and decoder $\mathcal{F}$, distilled dataset $\mathcal{D}^*$
(or original dataset $\mathcal{O}$ and classical dataset distillation algorithm $Alg$)
**Output:** Synthetic dataset $\mathcal{S}$
**Params:** Class $c$, classifier-free guidance (cfg) factor w, semantic guidance factor $\gamma$, guided
        range $t_l, t_h$, diffusion steps $t_f, t_r$, scales $\{\alpha_t\}_{t=1}^{t_f}$

1 Obtain the distilled dataset $\mathcal{D} = \mathcal{D}^*$ **if** $\mathcal{D}^*$ exists **else** $\mathcal{D} = Alg(\mathcal{O})$;
2 **for** *each image* $x_0 \in \mathcal{D}$ **do**
3     Perform ***Semantic Inheritance:*** $z_0 = \mathcal{E}(x_0)$;
4     Sample Gaussian noise: $\epsilon \sim \mathcal{N}(0, I)$;
5     Perform the forward process as Equation 7: $\hat{z}_{t_r} = z_{t_f} = \sqrt{\alpha_{t_f}} z_0 + \sqrt{1 - \alpha_{t_f}} \epsilon$;
6     **for** $t = t_r$ ***down to*** $1$ **do**
7         Obtain the predicted cfg noise: $\epsilon_\theta = \epsilon_\theta(\hat{z}_t, t, \varnothing) + \omega \cdot (\epsilon_\theta(\hat{z}_t, t, c) - \epsilon_\theta(\hat{z}_t, t, \varnothing))$;
8         **if** $t_l \leq t \leq t_h$ **then** *(Semantic Fusion)*
9             Compute the guidance as Equation 8: $\nabla_{\hat{z}_t} \mathcal{G}_t(\hat{z}_t) = (\hat{z}_t - z_0)\sigma_t$;
10            Execute ***Semantic Guidance:*** $\hat{z}_{t-1} = s(\hat{z}_t, t, \epsilon_\theta) - \gamma \cdot \nabla_{\hat{z}_t} \mathcal{G}_t(\hat{z}_t)$;
11         **else**
12            Execute vanilla sampling: $\hat{z}_{t-1} = s(\hat{z}_t, t, \epsilon_\theta)$;
13 **return** Decoded synthetic dataset: $\mathcal{S} = \{\mathcal{F}(\hat{z}_0)\}$;

---

## A.1 USE OF LLMS

We only use LLMs to do some minor grammar corrections and polishing of the manuscript.

## A.2 RELATED WORK

### A.2.1 DATASET DISTILLATION

Dataset Distillation (DD) condenses a large original dataset into a compact distilled dataset, preserving essential information to maintain comparable test performance when training models. These informative images are also valuable for various applications, including continual learning Yang et al. (2023); Gu et al. (2023), federated learning Yan et al. (2025); Huang et al. (2024), and neural architecture search Ding et al. (2024); Such et al. (2020). The classical DD framework utilizes bi-level optimization for dataset generation. This involves an inner loop that updates the network to assess classification performance, while the outer loop synthesizes images through specific matching strategies. Various DD approaches have emerged from this framework, including gradient matching Loo et al. (2023); Wang et al. (2023), distribution matching Wang et al. (2022a); Deng et al. (2024); Zhang et al. (2024), and trajectory matching Du et al. (2023); Zhong et al. (2025); Guo et al. (2023). Inspired by DFKD Liu et al. (2024); Chen et al. (2019), recent studies propose decoupled dual-time optimization approaches Yin et al. (2023); Shao et al. (2024) that separate the distillation process into synthesis and training phases, enabling more efficient handling of large datasets. The methods alleviate some constraints on visual semantics in distilled images.

Although the images synthesized by traditional methods are insightful and these expressiveness may enhance distillation performance, all perform the optimization process directly in the pixel space, which tends to excessively learn specific patterns that overfit on a prior architecture Cazenavette et al. (2023). The resulting distilled images exhibit *abstract, noisy*, and *unrealistic* characteristics. The visual semantics with realism that help generalize across architectures are suppressed Sun et al. (2024). Therefore, generation-based methods emerged.

### A.2.2 DATASET SYNTHESIS WITH GENERATIVE MODEL

Generative prior methods Cazenavette et al. (2023); Zhong et al. (2024) use GANs Karras et al. (2019; 2020) to synthesize images, transitioning the optimization space from pixels to latent codes. But they still rely on the traditional DD paradigm and their effectiveness remains constrained by expensive inner loop matching mechanisms and inadequate semi-realistic representation. Consequently, several diffusion-based approaches Su et al. (2024); Gu et al. (2024); Chan-Santiago et al. (2025); Chen et al. (2025); Zhao et al. (2025) leveraging the powerful synthesis capabilities of diffusion models have begun to emerge and demonstrated promising performance. These methods mark a radical break from the traditional DD paradigm, completely abandoning its established norms in order to refocus squarely on the core principles of coreset construction. Some novel strategies (such as prototype learning, influence function, etc.) are used to synthesize representative samples.

Inspired by both single-stage classical DD and generative-based approaches, we implement our work by integrating the expressive knowledge distilled from traditional DD with the powerful realistic image generation capability of diffusion models. This hybrid framework aims to synthesize a visually and semantically enriched dataset, thereby enhancing cross-architecture generalization.

### A.2.3 IMAGE RESTORATION WITH GENERATIVE MODEL

Image restoration is a fundamental discipline within image processing, concerned with the inverse problem of reconstructing a high-quality from its degraded observation. The field encompasses several key sub-domains, each targeting specific types of degradation, such as image deblurring Chen et al. (2024), super-resolution Yue et al. (2025), and adverse weather removal Peng et al. (2025) including tasks like de-raining, de-snowing, and de-hazing to eliminate weather-related artifacts.

Recently, some image restoration techniques Yue et al. (2025); Wu et al. (2024); Wang et al. (2024) based on diffusion models have begun to attract researchers' attention. Although similar to DIVER, both aim to synthesize realistic and semantically clear images using prior knowledge from generative models, they are fundamentally different. (1) Restoration tasks typically require ground-truth data (clean images) for training. In contrast, our method does not rely on any other images and does not require training. (2) Restoration seeks to recover a clean natural image from a degraded observation (e.g., blurring, noise, low resolution). Our distilled images are not degraded natural images, but optimization artifacts containing architecture-specific patterns, so the objective is semantic refinement rather than reconstruction.

Table 10: Distillation performance on the ConvNet architecture.

| Distill. Alg. | IPC | Distill. Mode | Fruit | Woof | Meow | Squawk | Nette | Yellow |
|---|---|---|---|---|---|---|---|---|
| DM | 1 | DD | $19.5_{\pm1.0}$ | $20.2_{\pm0.6}$ | $18.3_{\pm1.7}$ | $27.3_{\pm0.8}$ | $28.9_{\pm1.5}$ | $32.9_{\pm1.6}$ |
| | | GLaD | $18.9_{\pm1.7}$ | $18.2_{\pm1.9}$ | $20.5_{\pm2.1}$ | $23.2_{\pm1.6}$ | $30.3_{\pm1.1}$ | $30.3_{\pm1.0}$ |
| | | DIVER | $\mathbf{13.2}_{\pm1.0}$ | $\mathbf{13.6}_{\pm0.6}$ | $\mathbf{10.5}_{\pm1.3}$ | $\mathbf{18.2}_{\pm0.7}$ | $\mathbf{21.6}_{\pm1.2}$ | $\mathbf{17.8}_{\pm0.7}$ |
| | 10 | DD | $26.2_{\pm0.5}$ | $24.0_{\pm0.6}$ | $23.2_{\pm0.8}$ | $32.7_{\pm1.2}$ | $40.0_{\pm1.1}$ | $41.5_{\pm0.6}$ |
| | | DIVER | $\mathbf{18.4}_{\pm0.7}$ | $\mathbf{23.1}_{\pm0.8}$ | $\mathbf{19.6}_{\pm0.9}$ | $\mathbf{31.2}_{\pm1.7}$ | $\mathbf{35.0}_{\pm1.7}$ | $\mathbf{32.0}_{\pm0.4}$ |
| MTT | 1 | DD | $24.2_{\pm0.7}$ | $27.3_{\pm1.0}$ | $27.8_{\pm1.1}$ | $12.0_{\pm1.6}$ | $17.7_{\pm1.9}$ | $40.7_{\pm2.9}$ |
| | | GLaD | $21.4_{\pm1.2}$ | $25.6_{\pm1.4}$ | $23.5_{\pm1.8}$ | $13.6_{\pm1.2}$ | $22.5_{\pm1.4}$ | $33.9_{\pm1.4}$ |
| | | DIVER | $\mathbf{18.0}_{\pm0.4}$ | $\mathbf{18.7}_{\pm1.5}$ | $\mathbf{11.8}_{\pm0.9}$ | $\mathbf{11.6}_{\pm1.0}$ | $\mathbf{20.3}_{\pm1.5}$ | $\mathbf{19.4}_{\pm0.6}$ |
| | 10 | DD | $35.3_{\pm1.4}$ | $32.9_{\pm0.8}$ | $37.1_{\pm1.3}$ | $49.9_{\pm1.8}$ | $20.7_{\pm1.8}$ | $53.8_{\pm0.8}$ |
| | | DIVER | $\mathbf{25.7}_{\pm0.8}$ | $\mathbf{25.5}_{\pm0.8}$ | $\mathbf{28.3}_{\pm1.1}$ | $\mathbf{39.1}_{\pm1.0}$ | $\mathbf{34.3}_{\pm1.4}$ | $\mathbf{37.5}_{\pm1.3}$ |

## A.3 MORE EXPERIMENTS

**Quantitative Results on ConvNet.** As shown in Tab. 10, we compare the distillation performance of DM and MTT on ConvNet under DD and DIVER. In most cases, the performance of our method does drop significantly. We have analyzed the reasons in the text. This performance decline primarily stems from two factors: (1) The encoder maps the distilled images into a deep latent space, filtering out specific patterns (ConvNet) while preserving high-level semantics. This is evidenced by our experimental finding that directly decoding these latent representations (without employing the denoising model) still yields considerable generalization gains. (2) The diffusion model further eliminates residual low-level information in the latent space through denoising steps, thereby injecting additional class-specific information to enhance generalization. Notably, DIVER with MTT still achieve performance gains on the ImageNette dataset. This may be because the distilled images contain excessive invalid "noise", which not only suppresses semantic expression but also hinders distillation performance. In contrast, diffusion filters out this portion of ineffective low-level information, thereby improving distillation performance. This observation also inspires our future work, extracting meaningful low-level features to further improve distillation performance on ConvNet.

**Effect of Different Resolutions.** We compare the effect of DiT models with different resolutions on cross-architecture generalization in Tab. 12. As the resolution increases, the performance improves slightly, which may be because the synthesized images better inherit the semantic details of the distilled images. However, as shown in Tab. 13, this is accompanied by a significant increase in synthesis time. And better equipment can also improve the synthesis efficiency.

**Synthetic Images for Generative Models**. As shown in Tab. 11, we apply the synthetic dataset generated by MGD[3] directly to our framework, which reduces performance by approximately 10% (51.7% vs. 60.3%). Because the synthetic images have contained relatively clear semantic structures. When they are used in our framework, the performance degradation is mainly attributed to (1) the encoding of VAE from high-dimensional to low-dimensional space itself loses information, and (2) uncertainty introduced by diffusion. Prototype-based methods retain valuable information from the original dataset or prototype, thereby achieving performance improvements.

Table 11: Integrate MGD[3] into DIVER in different ways. The results are obtained by averaging three experiments each on ResNet-18, ResNetAP-10 and ConvNet-6 on ImageNette with IPC 10.

| Method | Original | Synthesis-based | Prototype-based |
|---|---|---|---|
| MGD[3] | $60.3_{\pm0.3}$ | $51.7_{\pm0.5}$ | $\mathbf{62.8}_{\pm0.4}$ |

**Effect under high IPC settings**. In practical applications, we place greater emphasis on the generalizability of information contained within distilled datasets to facilitate their use in training or fine-tuning other architectures, rather than being confined to a specific architecture like ConvNet. As illustrated in Tab. 14, datasets distilled through classical DD exhibit unstable performance under 50 IPC and 100 IPC. Traditional DD struggles significantly, with no improvement in generalization as IPC increases, even showing degradation under ConvNet evaluation. In contrast, our method demonstrates consistent gains, validating the significance of capturing universal information from original images rather than architecture-specific features to enhance generalization.

Table 12: Effect of DiT models with different resolutions on cross-architecture generalization on ImageNet Subsets .

| Resolution | Fruit | Woof | Meow | Squawk | Nette | Yellow |
|---|---|---|---|---|---|---|
| 256×256 | 22.3 | 16.2 | 15.7 | 17.2 | 20.3 | 20.2 |
| 512×512 | 22.4 | 17.1 | 17.8 | 17.4 | 23.3 | 20.5 |

Table 13: The synthesis time and GPU memory cost of different devices on ImageNet.

| Device | Resolution | GPU (GB) $\downarrow$ | Times (s) $\downarrow$ | |
|---|---|---|---|---|
| | | | DiT | Ours |
| 4090 | 256×256 | 4.02 | 2.41 | 2.48 |
| | 512×512 | 5.44 | 12.84 | 13.08 |
| A800 | 256×256 | 4.02 | 1.92 | 1.97 |
| | 512×512 | 5.44 | 6.74 | 6.91 |

**Visualization**. As shown in Fig. 8, different images also exhibit distinct optimal noise-addition steps. For each image, as the number of forward steps increases, the feature variations (e.g., texture details) gradually intensify. However, this change eventually diminishes and stabilizes, as the initial latent variables asymptotically converge to the same Gaussian distribution. Our visual comparison of MTT and our DIVER under IPC=10 is presented in Fig. 9 ∼ Fig. 14. Our method releases the expressive semantics that are suppressed in raw DD, and the valuable label semantic information covers almost the entire image.

Table 14: Effect under high IPC settings.

| IPC | | 10 | 50 | 100 |
|---|---|---|---|---|
| ConvNet | DM | $23.2_{\pm0.8}$ | $22.7_{\pm1.1}$ | $20.9_{\pm2.1}$ |
| | Ours | $19.6_{\pm0.9}$ | $24.8_{\pm0.3}$ | $28.5_{\pm1.5}$ |
| CrossArc | DM | $17.6_{\pm1.7}$ | $16.4_{\pm1.5}$ | $16.8_{\pm1.3}$ |
| | Ours | $20.0_{\pm1.6}$ | $26.5_{\pm2.0}$ | $32.6_{\pm1.8}$ |

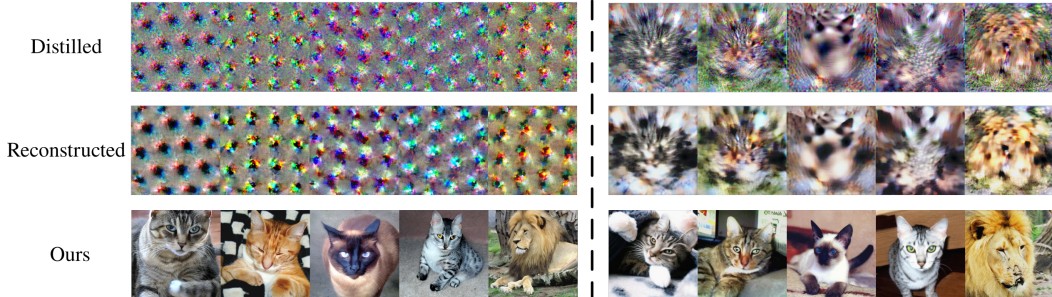

Figure 7: Comparison of distilled images, reconstructed images (obtained through direct VAE encoding and decoding without DiT), and our synthetic images with DM(left) and MTT(right).

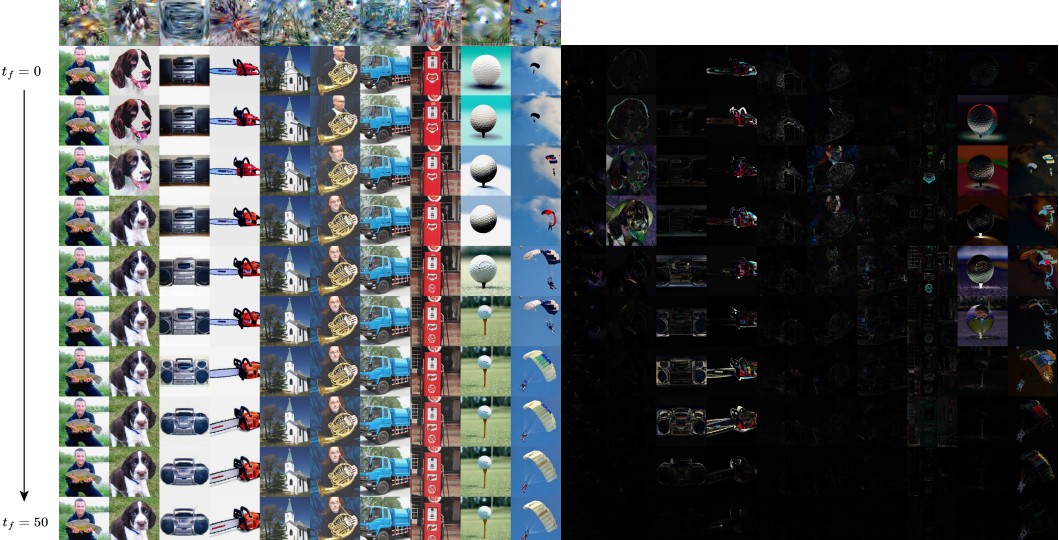

Figure 8: The pixel difference between the images synthesized by adjacent forward steps.

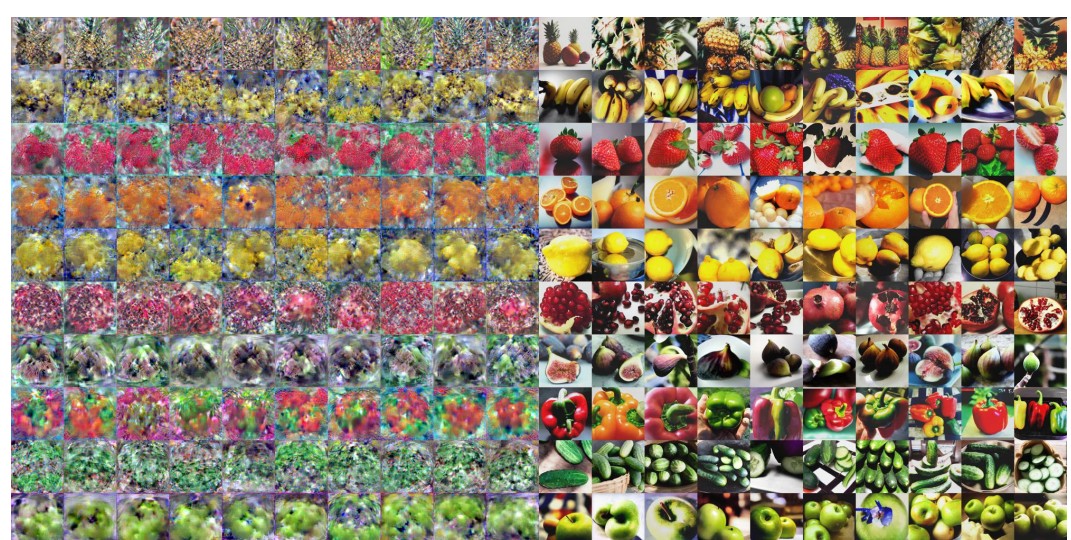

Figure 9: Visualizaiton comparison between raw DD and DIVER on ImageFruit.

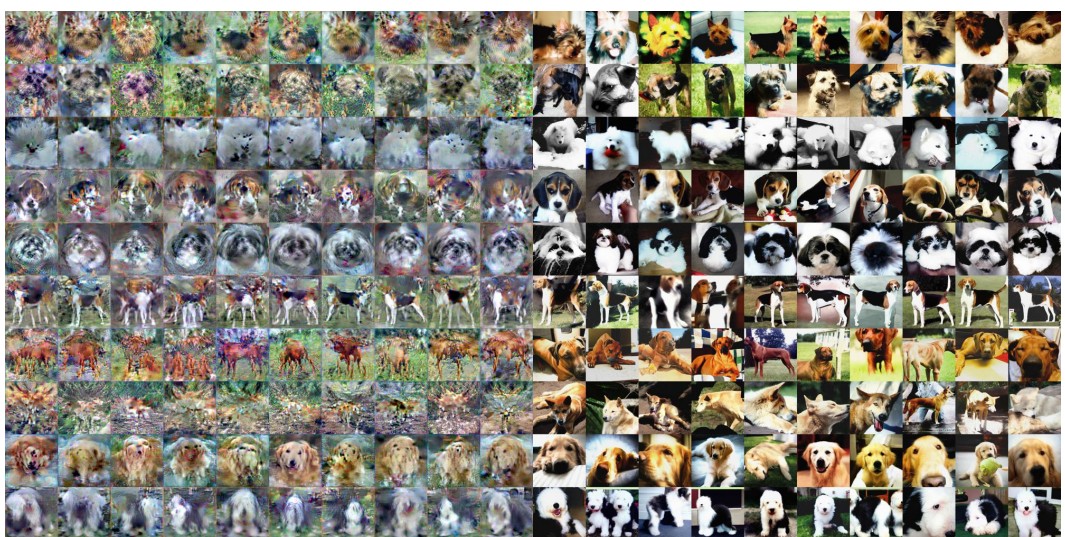

Figure 10: Visualizaiton comparison between raw DD and DIVER on ImageWoof.

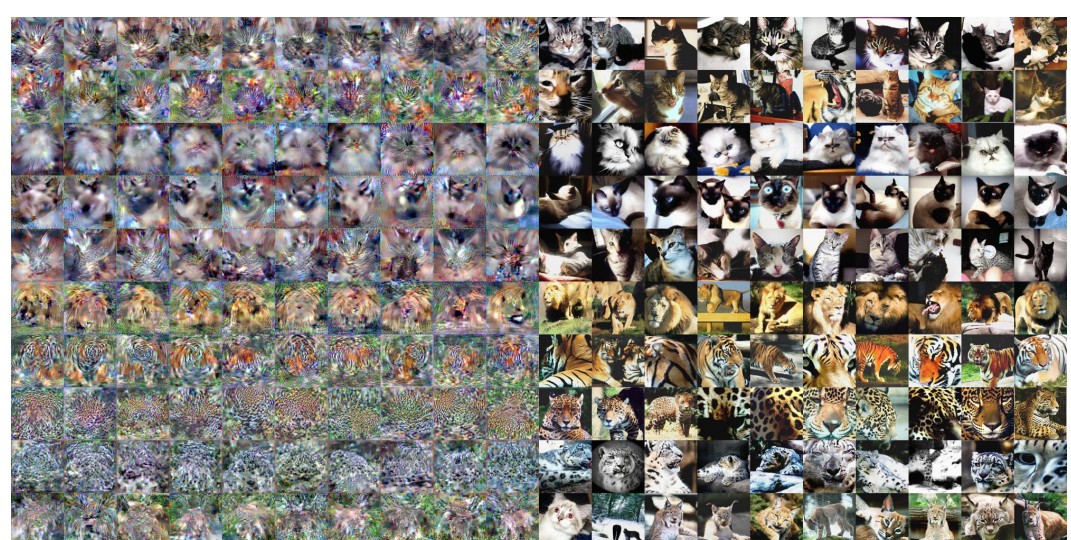

Figure 11: Visualizaiton comparison between raw DD and DIVER on ImageMeow.

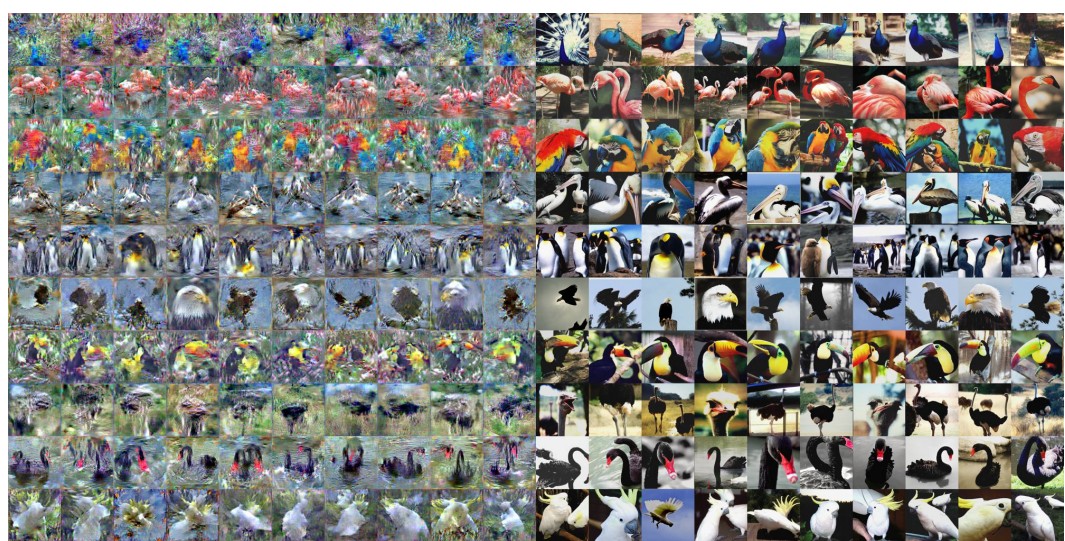

Figure 12: Visualizaiton comparison between raw DD and DIVER on ImageSquawk.

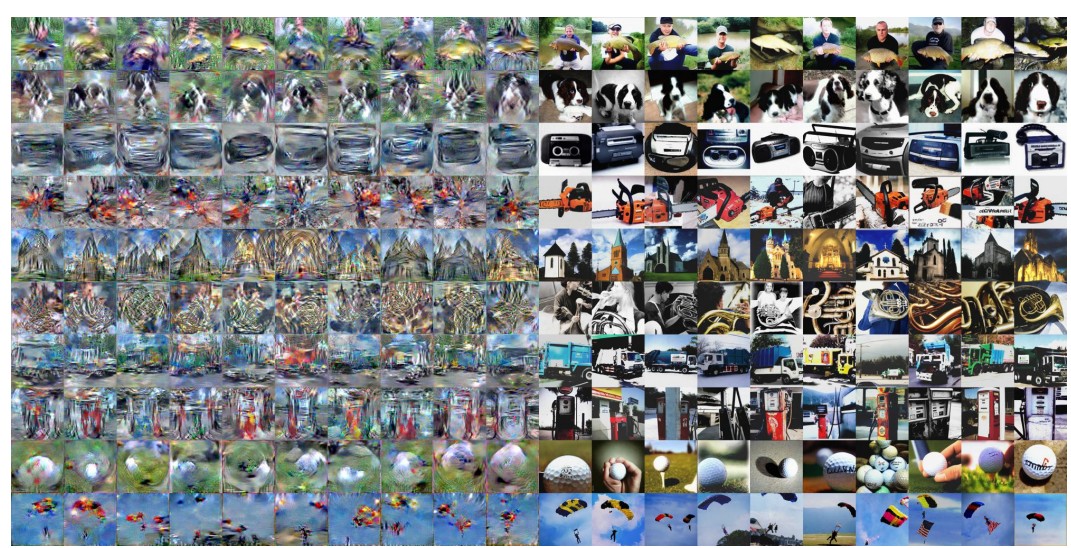

Figure 13: Visualizaiton comparison between raw DD and DIVER on ImageNette.

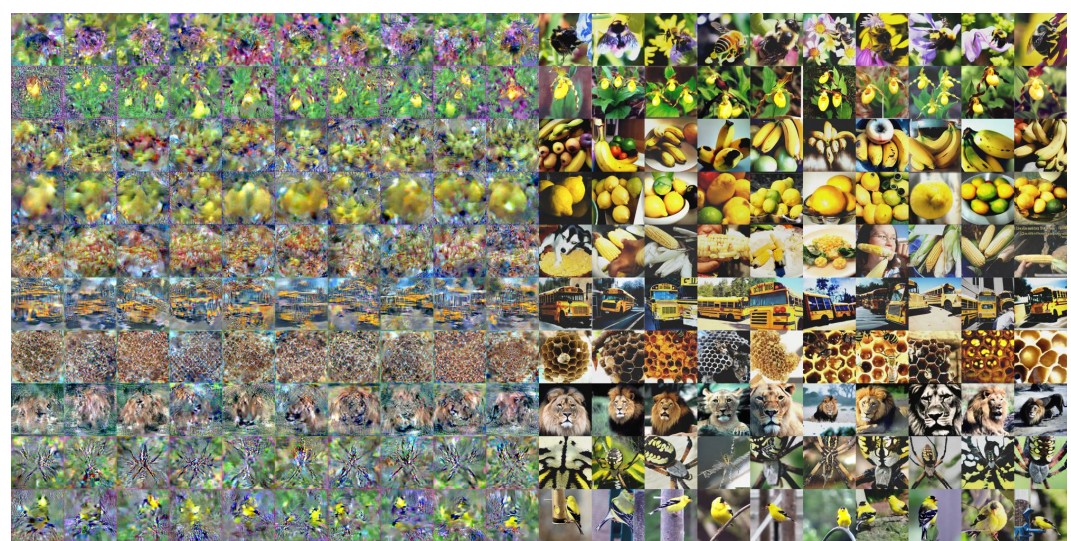

Figure 14: Visualizaiton comparison between raw DD and DIVER on ImageYellow.

