# OpenReview forum: "DIVER: Diving Deeper into Distilled Data via Expressive Semantic Recovery"
_ICLR.cc/2026/Conference — Submitted to ICLR 2026_

### Official Review · Reviewer_D6Z8 · 2025-10-28

**Soundness:** 3
**Presentation:** 1
**Contribution:** 2
**Rating:** 4
**Confidence:** 4

**Summary:**

This paper addresses the problem of cross-architecture generalization in dataset distillation. Unlike prior approaches that generate synthetic images directly with generative models, the authors propose using generative models to refine already distilled images. Specifically, given a distilled image, the method embeds it into the latent space of a diffusion model, perturbs the latent embedding with noise, and then denoises it using the diffusion model combined with a guidance technique that constrains the output to remain close to the original embedding. Experimental results demonstrate that this refinement process improves the cross-architecture generalization performance of existing dataset distillation methods.

**Strengths:**

* The paper introduces a novel framework that refines distilled images to improve cross-architecture generalization. This approach has the potential to open a promising research direction and inspire further studies.
* The proposed method is simple yet effective.

**Weaknesses:**

* The rationale behind why the proposed method works remains unclear. My understanding is that the method makes distilled images appear more realistic while preserving their overall structure and texture. This could explain improvements for unrealistic images, but the reported gains also extend to already realistic distilled images (e.g., $D^4M$ and $MGD^3$), which is less intuitive. The authors briefly state that the method "stably optimizes the prototypes," but this explanation lacks clarity.

* The presentation can be improved:
  * Eq. (1) is not clearly explained. The right-hand side deviates from the typical formulation of dataset distillation, where the goal is usually to synthesize a dataset such that models trained on it achieve low empirical loss on the original dataset. Matching loss values alone seems insufficient, since the objective could be met even if both losses are high but similar. In addition, if $\Phi_O^p$ and $\Phi_D^p$ are identical, it is unclear why they are written differently. The definitions of $L_L$ and $H_L$ are also abstract and ambiguous, and their absence on the right-hand side further reduces the clarity of the equation.
  * Similarly, $h$ and $\Delta R$ in Eq. (5) are introduced in an abstract manner. They also seem specific to the proposed method rather than the general framework for refining synthetic images, which may cause confusion.
  * Several expressions throughout the paper are vague or difficult to interpret, such as: "releases the visual semantics" (line 52), "lie dormant in the deep ~ specific artifacts" (lines 81–82), "characterizes the visual-semantic authenticity gap" (line 193), "Subsequently, in CP, limited ~ synthetic images" (lines 252–254), and "stably optimize the prototypes" (line 353).

* The method also appears related to restoration techniques based on diffusion models (e.g, [1]). Discussing these connections would strengthen the contribution.

[1] Arbitrary-steps Image Super-resolution via Diffusion Inversion, CVPR 2025.

**Questions:**

* Why does the proposed method improve the performance of already realistic images?
* Are the reported results averaged over five different architectures, with each architecture evaluated five times?
* In Table 6, was CFG used for the fourth row?
* The paper mentions that the VAE in latent diffusion models itself enhances generalization. Could you provide quantitative results to support this claim?

---

> ### Author Response · Authors · 2025-11-23
> **Response to Reviewer D6Z8 (1/2)**
>
> We thank the reviewer for the insightful comments and constructive suggestions, which have helped us improve the manuscript. We have revised the manuscript accordingly and provide our point-by-point responses below.
>
> **Q1 (@Weakness 1 and Question 1)：Why does the proposed method improve the performance of already realistic images?**
>
> We sincerely apologize for the misunderstanding caused by our unclear expression. We do not apply images synthesized by D$^4$M or MGD$^3$ containing real semantic information to our entire framework. Instead, we adhere to their criteria and encode the full dataset into the latent space for prototype learning (k-means), and then use SG and SF to preserve prototype information to the greatest extent possible for data synthesis.
>
> Indeed, we find that directly applying DIVER to realistic synthetic data results in a significant performance degradation, which is intuitively reasonable. This is mainly due to (1) the encoding of VAE from high-dimensional to low-dimensional space itself loses information , and (2) uncertainty introduced by diffusion. Through experiments on the ImageNette under 10 IPC (the results are obtained by averaging three experiments each on ResNet-18, ResNetAP-10 and ConvNet-6), we observe that directly applying DIVER to real synthetic images result in a performance loss of nearly 10% (51.7% vs. 60.3%).
>
> | Method           | Original | Synthesis-based | Prototype-based |
> | :---------------: | :------: | :--------------: | :--------------: |
> | MGD$^3$ | 60.3±0.3 | 51.7±0.5        | **62.8±0.4**    |
>
> We have provided the more detailed description in the updated paper (lines 358-369) on how DIVER is applied to diffusion-based approaches.
>
> **Q2 (@Weakness 2)：The presentation can be improved.**
>
> Thanks for your valuable suggestions. Equation (1) is initially defined in a general form, primarily adopted from previous research [1,2].  We completely revise the presentation to conform to the typical formulation of dataset distillation.
>
> Specifically, we first point out that the initial formula for dataset distillation can be formulated as:
> $$
> \sup\limits_{x,\tilde{x}\sim P} \left|\mathcal{L}(\Phi_\mathcal{O}^p(x),y)-\mathcal{L}(\Phi_\mathcal{D}^p(\tilde{x}),\tilde{y})\right| \leqslant \epsilon
> $$
> This aims to build a compact distilled dataset $\mathcal{D}$ that extracts rich information from $\mathcal{O}$, such that models trained on $\mathcal{D}$ achieve performance within an acceptable deviation $\epsilon $ from those trained on original dataset $\mathcal{O}$. $\varPhi_\mathcal{O}^p$ and ${{\varPhi}}_{\mathcal{D}}^{p}$ are written differently because they can be derived from training with different initial parameters.
>
> To make this metric practically solvable, classical DD methods introduce ${\varPhi}$ to obtain informative guidance from $\mathcal{O}$ and $\mathcal{D}$ in a chosen representation space, and iteratively optimize $\mathcal{D}$ accordingly:
>
> $$
> \mathcal{D}^* = \arg\min_\mathcal{D} \mathcal{M}(\Phi_\mathcal{O}^p(x), \Phi_\mathcal{D}^p(\tilde{x}))
> $$
>
> where $\mathcal{M}$ represents various matching metrics, such as distribution matching, gradient matching, and trajectory matching.
>
> Then, we remove abstract expressions such as $L_L$, $H_L$, and $\varDelta R$, and instead define DDD in a more direct way:
>
> $$
> \mathcal{S}^* = \arg\min_\mathcal{S} \mathcal{M}(\Phi_\mathcal{O}^v(x), \Phi_\mathcal{S}^v(\mathcal{H_D*}(\tilde x)) \quad \text{s.t.} \mathcal{D}^* \leftarrow \mathcal{O}
> $$
>
> where $\mathcal{H}_{{D}^*}(\tilde x)$ is the synthetic image, the size $|\mathcal{S}|=|\mathcal{D}|  \ll |\mathcal{O}|$.   $\mathcal{H}$ denotes the semantic recovery strategies, which is designed to refine the semantics of distilled images, unlocking their suppressed potential for generalization.  Generally speaking, while the initial objective is designed for all $\varPhi$, its practical effectiveness is primarily observed at ${\varPhi}^{p}$. Our goal is to refine the distilled images to ensure its applicability at ${\varPhi}^{v}$ as well.
>
> We have revised some vague expressions in the paper, specifically:
>
> "releases the visual semantics"  $\rightarrow$ "As shown in Fig. 4, although this makes the distilled image appear to have some semi-clear semantic expression" （line 52）
>
> "lie dormant in the deep ~ specific artifacts" $\rightarrow$ “ this meaningful information is suppressed by spe-
> cific artifacts and unclear semantics, thus reducing generalization.” (lines 81-82)
>
> Remove "characterizes the visual-semantic authenticity gap".
>
> "stably optimize the prototypes"  $\rightarrow$ "D${}^{\text{4}}$M and MGD${}^{\text{3}}$ encode the full dataset into the latent space via a VAE for prototype learning,~" (lines 358-369)
>
> [1] MGD$^3$: Mode-Guided Dataset Distillation using Diffusion Models, ICML 2025.
>
> [2] Going Beyond Feature Similarity: Effective Dataset Distillation based on Class-aware Conditional Mutual Information, ICLR 2025.

---

> ### Author Response · Authors · 2025-11-23
> **Response to Reviewer D6Z8 (2/2)**
>
> **Q3 (@Weakness 3)：The method also appears related to restoration techniques based on diffusion models (e.g, [1]). Discussing these connections would strengthen the contribution.**
>
> Thank the reviewer for the insightful suggestions. In Section A.2.3 of the Related Work in our updated paper, we have analyzed the connection between diffusion-based image restoration and our DIVER, and cited the relevant literature on image restoration mentioned by the reviewer, specifically:
>
> They both aim to **synthesize realistic and semantically clear images using prior knowledge from generative models**, they are fundamentally different.
>
> (1) Restoration tasks typically require **ground-truth data (clean images) for training**. In contrast, our method **does not rely on any other images and does not require training.**
>
> (2) Restoration seeks to recover a clean **natural image** from a degraded observation (e.g., blurring, noise, low resolution). Our distilled images are not degraded natural images, but **optimized products that includes "noise" and artifacts** of architecture-specific patterns, so the objective is semantic recovery rather than reconstruction.
>
> **Q4 (@Question 2)：Are the reported results averaged over five different architectures, with each architecture evaluated five times?**
>
> Yes. We evaluate each architecture (five architectures in total) five times and calculate the mean and standard deviation.
>
> **Q5 (@Question 3)：In Table 6, was CFG used for the fourth row?**
>
> Yes. In all experiments involving DiT, we use CFG. We have updated Figure 2 to better understand the implementation of our entire process.
>
> **Q6 (@Question 4)：The paper mentions that the VAE in latent diffusion models itself enhances generalization. Could you provide quantitative results to support this claim?**
>
>  We have supplemented the revised version of our paper with experiments that employ only the VAE (without diffusion).  As shown in Table 9, the reconstructed images by VAE also enhance generalization to some extent. Furthermore, after semantic refinement with diffusion (Ours), the generalization performance improves even more. Our entire process demonstrated a systematic trade-off.  Further analysis and visualization results have been updated in Figure 7 and lines 459-467.
>
> | Method | Architecture | Distilled | Reconstructed |   Ours   |
> | ------ | :----------: | :-------: | :-----------: | :------: |
> | DM     |   ConvNet    | 23.2±0.8  |   21.2±1.2    | 19.6±0.9 |
> |        |   CrossArc   | 17.6±1.7  |   18.9±1.0    | 20.0±1.6 |
> | MTT    |   ConvNet    | 37.1±1.3  |   31.6±0.7    | 28.3±1.1 |
> |        |   CrossArc   | 16.1±1.4  |   21.8±1.9    | 26.7±1.7 |

---

> ### Author Response · Authors · 2025-11-27
> **A respectful request for discussion**
>
> Thanks a lot for your time in reviewing and insightful comments, according to which we have carefully revised the paper to answer the questions. We sincerely understand you’re busy. But since the discussion due is approaching, would you mind checking the response to confirm where you have any further questions?
>
> We are looking forward to your reply and happy to answer your further questions.
>
> Best regards
>
> Authors

---

> ### Comment · Reviewer_D6Z8 · 2025-11-27
>
> Thank you for the detailed response. The revisions have certainly improved the clarity of the manuscript. However, my primary concern remains that DIVER is not directly applicable to methods that produce already realistic images.
>
> Although the authors demonstrated that a modified version—replacing SI with a component of the original method while applying SG and SF—can improve performance, the necessity of such modification itself underscores the method's limited versatility. Furthermore, **without the SI component, the remaining SG and SF techniques bear a strong resemblance to those used in MGD$^3$**, which diminishes the technical novelty. Crucially, this modification strategy appears incompatible with other diffusion-based frameworks, such as IGD [1], which initializes latents with noise. Therefore, I remain unconvinced of the method's general applicability and distinct contribution, unless it is demonstrated that DIVER-enhanced images (from non-realistic baselines) can surpass the performance of methods that inherently produce realistic images.
>
> [1]: Influence-Guided Diffusion for Dataset Distillation, ICLR 2025

---

> ### Author Response · Authors · 2025-11-27
> **Further Discussion**
>
> Thank you for your positive reply! We fully understand your concerns regarding the general applicability of our method.  Below is a summary of our responses to your further questions:
>
> ● What we aim to clarify is that “without the SI component” means that semantic information is no longer inherited from the distilled images via the VAE, but it is **still inherited from the prototypes**, which is an aspect absent in $MGD^3$. We find that initializing the latent with prototype information, rather than random noise, indeed yields performance gains.
>
> ● We acknowledge that this modification strategy is difficult to extend to IGD. However, **the core focus of our work is on recovering the semantics of abstract images rather than on applications to realistic images**. Therefore, we do not expect it to apply to all diffusion-based methods, and its applicability to $MGD^3$ and $D^4M$ should be considered merely an additional contribution. On the other hand, although our method cannot be directly extended to IGD, the fusion of $MGD^3$ still surpasses the performance of IGD, achieving state-of-the-art (SOTA) performance, which is also of great significance.
>
> | IPC  | DiT+IGD    | MGD³+Ours | Minimax+ IGD | Minimax+MGD³+Ours |
> | ----  | :--------: | :--------: | :---------: | :-----------------: |
> | 10  | 45.5±0.5 | 46.4±0.3  | 46.2±0.2  | **46.9±0.5**      |
> | 50   | 59.8±0.3 | 60.6±0.4| 60.3±0.4  | **61.0±0.2**      |
>
> ● Classical DD and diffusion-based DD represent two distinct branches of research. Due to inherent limitations in classical DD, comparing them directly may be considered inequitable. **Recent studies on classical DD (such as NCFM(CVPR 2025) [1], EDF (CVPR 2025)[2], HDD (NeurIPS 2025) [3], etc.) have not included diffusion-based methods as baselines (a decision that holds its own research significance)**. As previously mentioned, our primary objective is to recovery the semantics of abstract images synthesized by classical DD, rather than handling realistic images generated via diffusion-based approaches. Therefore, we respectfully emphasize **the difficulty of supplementing such a benchmark [DIVER-enhanced images (from non-realistic baselines) can surpass the performance of methods that inherently produce realistic images] at this stage**, as well as **its divergence from the focus of our submission**.
>
> [1] Dataset Distillation with Neural Characteristic Function: A Minmax Perspective, CVPR 2025.
>
> [2] Emphasizing Discriminative Features for Dataset Distillation in Complex Scenarios, CVPR 2025.
>
> [3] Hyperbolic Dataset Distillation, NeurIPS 2025.

---

### Official Review · Reviewer_QTpw · 2025-10-29

**Soundness:** 3
**Presentation:** 3
**Contribution:** 3
**Rating:** 4
**Confidence:** 5

**Summary:**

The paper proposes DIVER, a two-stage dataset distillation framework that enhances the meaningful semantics of distilled datasets. The first stage follows the classical DD method to obtain a distilled dataset, and the second stage DDD leverages a pre-trained diffusion model with three semantic recovery strategies. DIVER operates in a training-free and raw-data-free manner, acting as a plug-in refinement to existing DD datasets. Experiments across multiple ImageNet subsets and architectures show improved cross-architecture generalization over classical and diffusion-based DD methods with minimal computational overhead.

**Strengths:**

- **Novelty**: The idea of performing a second-stage refinement on top of the distilled data is conceptually new and interesting. The idea is similar to the denoising process of diffusion models: diffusion models aim to reconstruct images from noise, and DIVER aims to recover semantics from the distillation dataset.
- **Practicality**: The proposed method is plug-and-play and does not require re-training or access to the original data, which is desirable for privacy-preserving.
- **Clear motivation**: The authors point out that many DD methods overfit to the architecture used for distillation, leading to poor cross-architecture generalization.
- **Efficiency**: The method demonstrates low GPU memory usage (≈4 GB) and comparable sampling time on DiT, which strengthens its practical relevance.

**Weaknesses:**

- **Limited Mathematical Depth**: Although the paper introduces several intuitively motivated components (semantic inheritance, guidance, and fusion), the overall formulation lacks mathematical rigor and theoretical grounding. The proposed strategies are primarily described at the conceptual level, with limited formal analysis of their effect on the data distribution, diffusion dynamics, or semantic representation. In particular, the core idea of separating architecture-specific and content semantic information is presented heuristically without a clear probabilistic or optimization-based framework. As a result, the paper is more like an empirical design than a theoretically grounded contribution.
- **Lack of strong SOTA results**: While DIVER demonstrates consistent improvements over several distillation methods, the reported gains do not surpass the current state-of-the-art diffusion-based or generative prior methods by a convincing margin (for example, can you provide IGD + DIVER or MGD3 + DIVER on ImageNet-1k? So as the other datasets.).
- Overall, I think DIVER is an interesting work and I would like to raise my score if the authors can address all my concerns and questions.

**Questions:**

- **About SI**: SI relies on a pre-trained VAE encoder trained on real images. However, the distribution of distilled data is highly non-natural and abstract, often lying far outside the real-image manifold. How can you ensure that this encoder can successfully separate architecture-specific artifacts from genuine semantic content in such an out-of-distribution setting? Without explicit mathematical disentanglement or domain adaptation, the claim that SI can “filter out architecture-specific noise” seems unsubstantiated.
- **About SG**: The guidance defined as the distance between $z_0$ and $z_t$ is a heuristic design. The authors argue that latent codes suffer from information degradation during denoising,  but $z_0$ is derived from the encoded distilled image, which does not contain a clear semantic structure according to the visualizations in Fig. 2. Therefore, it is unclear what semantic information this guidance preserves. The motivation and theoretical foundation for SG require much clearer justification.
- **About SF**: The authors claim that SF injects label semantics only during the “semantic phase” of denoising. However, according to the supplementary material, label prompts (or label ID) are provided throughout the denoising process, as required by the DiT’s classifier-free guidance. If so, SF is not a new methodological component but an inherent property of the diffusion model itself. Could the authors clarify how SF differs in practice from the standard DiT conditioning process?
- How sensitive is the performance to the choice of diffusion model? Have you tried different pre-trained models (e.g., SD v1.5)?
- What is the purpose of applying DIVER to the generative DD method, except for performance gain? Since the synthetic data of the generative model already contains clear semantic information.
- Why you argue that the generative DD methods "going so far as to fully discard its conventions in favor of strictly returning to the fundamentals of coreset selection"? (line 75-77)

---

> ### Author Response · Authors · 2025-11-23
> **Response to Reviewer QTpw (1/2)**
>
> Thank you very much for your constructive feedback, the valuable question and your interest in DIVER. This has given us a deeper understanding of DIVER. We answer your questions as below.
>
> **Q1 (@Weakness 1, Question 1 and  Question 2 )：Limited Mathematical Depth, About SI and SG**
>
>  As noted, distilled data largely lies outside the real-image manifold, and its semantics lack a quantitative evaluation standard. Therefore, it is challenging to use comprehensive theoretical mathematical formulas to explain. We primarily leverages the inherent theoretical properties of VAE and guided diffusion to explain our overall pipeline.
>
> (1) The VAE compresses high-dimensional images into a lower-dimensional latent space, inherently preserving semantic content. Latent code is essentially a compressed structured and semantic representation [1].
>
> (2) As described in Section 2.2, our guided diffusion process (energy-based models, EBM) [2]  is adaptively steered by learned feedback mechanisms (specifically, Semantic Guidance related to z$_0$) to generate target-specific outputs according to user requirements.
>
> Regarding (1), we first visualize images reconstructed directly by the VAE (without DiT)  in Figure 7. These reconstructed images are notably cleaner, confirming that the VAE effectively filters out "noise" . We further evaluate these reconstructed images in Table 9, observing improved cross-architecture performance alongside a decline in ConvNet performance. This verifies that the filtered "noise" is indeed architecture-specific noise pertinent to ConvNet.
>
> | Method | Architecture | Distilled | Reconstructed | Ours     |
> | ------ | :------------: | :---------: | :-------------: | :--------: |
> | DM     | ConvNet      | 23.2±0.8  | 21.2±1.2      | 19.6±0.9 |
> |        | CrossArc     | 17.6±1.7  | 18.9±1.0      | 20.0±1.6 |
> | MTT    | ConvNet      | 37.1±1.3  | 31.6±0.7      | 28.3±1.1 |
> |        | CrossArc     | 16.1±1.4  | 21.8±1.9      | 26.7±1.7 |
>
> Regarding (2), our objective is to generate semantically clearer images while retaining the essential information from the extracted z$_0$. To this end, we apply SG during guided diffusion.  We visualize the distribution of various images in a low-dimensional space using t-SNE in Figure 5. We observe that the distilled images form a very compact cluster, significantly displaced from the original image distribution. However, our SG not only pulls the distribution of our method (DIVER) closer to the natural manifold while bringing the overall distribution closer to the distilled dataset, thereby enhancing diversity and  generalization capability.  We believe this is a trade-off between preserving information from the distilled dataset and recovering clear semantics (original data distribution).
>
> [1] An Introduction to Variational Autoencoders. Foundations and Trends in Machine Learning, 2019.
>
> [2] Universal Guidance for Diffusion Models. CVPR, 2023.
>
>
> **Q2 (@Question 3 )：About SF: The authors claim that SF injects label semantics only during the “semantic phase” of denoising. However, according to the supplementary material, label prompts (or label ID) are provided throughout the denoising process. Could the authors clarify how SF differs in practice from the standard DiT conditioning process?**
>
> We sincerely apologize for any confusion and misunderstanding caused by our unclear expression.  We employ Classifier-Free Guidance (CFG) to inject label semantics at each denoising stage. Semantic fusion is essentially a "specific phase" (semantic phase ) where we integrate inherited semantics, label semantics, and those guided semantics by z$_0$. Other stages remain unguided to preserve semantic clarity. Since the semantics of abstract data are inherently ambiguous, full-stage guidance would lead to visual artifacts shown in Figure 4 (right), and performance degradation reported in Table 7.
>
> We have updated our semantic evolution process (Figure 2), along with the relevant description (lines 250-258) and pseudocode for clearer understanding.
>
> **Q3 (@Weakness 2)：Lack of strong SOTA results: the reported gains do not surpass the current state-of-the-art diffusion-based or generative prior methods by a convincing margin (for example, can you provide IGD + DIVER or MGD3 + DIVER on ImageNet-1k? So as the other datasets.).**
>
> In the updated version of the paper (Table 4), we apply our method to MGD$^3$and achieve state-of-the-art results through original DiT and Minimax fine-tuning on ImageNet-1K.
>
> | IPC  | SRe²L    | G-VBSM   | RDED     | DiT      | Minimax  | D⁴M      | MGD³     | MGD³+Ours | MGD³+Minimax+Ours |
> | ---- | :--------: | :--------: | :--------: | :--------: | :--------: | :--------: | :--------: | :---------: | :-----------------: |
> | 10   | 21.3±0.6 | 31.4±0.5 | 42.0±0.1 | 39.6±0.4 | 44.3±0.5 | 27.9±0.2 | 45.8±0.3 | 46.4±0.3  | **46.9±0.5**      |
> | 50   | 46.8±0.2 | 51.8±0.4 | 56.5±0.1 | 52.9±0.6 | 58.6±0.3 | 55.2±0.1 | 60.2±0.1 | 60.6±0.4  | **61.0±0.2**      |

---

> ### Author Response · Authors · 2025-11-23
> **Response to Reviewer QTpw (2/2)**
>
> **Q4 (@Question 4)：How sensitive is the performance to the choice of diffusion model? Have you tried different pre-trained models (e.g., SD v1.5)?**
>
> Thank you for your insightful question. We test different generative models, including the diffusion-based SD-V1.5 and the flow-based SiT (providing flow guidance (z$_0$-z$_t$) at 0.2 < t < 0.6), both showing consistent performance improvements. The smaller gain with SD-V1.5 is attributed to architectural discrepancies (U-Net vs. Transformer) and its different pre-training dataset (not ImageNet-1K).
>
> | IPC  |   DD   | Ours (SD-V1.5) | Ours (DiT) | Ours (SiT) |
> | ---- | :--------: | :--------------: | :----------: | :----------: |
> | 1    | 17.7±1.9 | 19.1±1.3       | 20.3±1.5   | 20.2±1.7   |
> | 10   | 20.7±1.8 | 26.5±2.1       | 34.3±1.4   | 33.1±2.3   |
>
> **Q5 (@Question 5)：What is the purpose of applying DIVER to the generative DD method, except for performance gain? Since the synthetic data of the generative model already contains clear semantic information.**
>
> We sincerely apologize for any misunderstanding. We do not apply images synthesized by D$^4$M or MGD$^3$ containing real semantic information to our entire framework. Instead, we encode the full dataset into the latent space according to their criterion for prototype learning, and then use SG and SF to preserve prototype information to the greatest extent possible for data synthesis.
>
> We find that directly applying DIVER to realistic synthetic data results in a significant performance degradation, which is intuitively reasonable. This is mainly due to:
>
>  (1) the encoding of VAE from high-dimensional to low-dimensional space itself loses information.
>
>  (2) uncertainty introduced by diffusion.
>
> Through experiments on the ImageNette under 10 IPC (the results are obtained by averaging three experiments each on ResNet-18, ResNetAP-10 and ConvNet-6), we observe that directly applying DIVER to real synthetic images (synthesis-based) result in a performance loss of nearly 10% (51.7% vs. 60.3%).
>
> | Method           | Original | Synthesis-based | Prototype-based |
> | --------------- | :-------: | :-------------: | :--------------: |
> | MGD$^3$ | 60.3±0.3 | 51.7±0.5        | **62.8±0.4**    |
>
> As for other purposes, we can leverage our framework to use diffusion and distilled datasets for generating adversarial examples [1] or evasion attacks [2], which is also of practical significance in the future.
>
> [1] Diffusion Models for Imperceptible and Transferable Adversarial Attack, TPAMI 2024.
>
> [2] DiffAttack: Evasion Attacks Against Diffusion-Based Adversarial Purification, NeurIPS 2023.
>
> **Q6 (@Question 6)：Why you argue that the generative DD methods "going so far as to fully discard its conventions in favor of strictly returning to the fundamentals of coreset selection"? (line 75-77)**
>
> Because diffusion-based DD (such as D$^4$M, MGD$^3$, and even Minimax) utilizes k-means prototype learning and the minimax criterion to generate representative or diverse samples, this is similar to the principles of earlier coreset-based methods. However, classic DD, such as trajectory-based or distribution-matching methods, have been gradually abandoned due to generalization bottlenecks. We believe that traditional DD still have room for development, but have been overlooked. Therefore, we begin this work, hoping to evolve our two-stage method into a single-stage method in the future to generate semantically clear synthetic images while satisfying traditional matching criteria. (For example, matching criteria could be directly used to guide the sampling stage of diffusion.)

---

> ### Author Response · Authors · 2025-11-27
> **A respectful request for discussion**
>
> Thanks a lot for your time in reviewing and insightful comments, according to which we have carefully revised the paper to answer the questions. We sincerely understand you’re busy. But since the discussion due is approaching, would you mind checking the response to confirm where you have any further questions?
>
> We are looking forward to your reply and happy to answer your further questions.
>
> Best regards
>
> Authors

---

> > ### Comment · Reviewer_QTpw · 2025-11-27
> >
> > Thank you for addressing some of my concerns. Here are some additional questions:
> > 1. According to the y-axis of Figure 2, the semantic information of the DD image is low at the beginning. The semantic begins to increase only during the denoising process. Thus, my main concern with SI is that, since you have named the process semantic inheritance, you should first clarify whether there is meaningful semantic information in the distilled image to be inherited. Since the distilled images get a different distribution from the raw images [R1], the output embeddings of the VAE encoder from an unseen distribution might be invalid for the denoising process. And the increased semantic information can be attributed to the pretrained diffusion. I do not mean that the distilled images are useless, however, the contribution of SI should be explained more clearly.
> >
> > 2. (about SI) Have you considered the alternative method like DDIM inversion?
> >
> > [R1]: Yang W, Zhu Y, Deng Z, et al. What is Dataset Distillation Learning?[C]//International Conference on Machine Learning. PMLR, 2024: 56812-56834.

---

> > > ### Author Response · Authors · 2025-11-28
> > > **Further Discussion**
> > >
> > > We appreciate that your additional questions and comments have not only enhanced the quality of our manuscript but have also prompted further investigation into the SI. Here are our replies:
> > >
> > > 1. Initially, we briefly stated in the introduction that distilled images contain meaningful information (lines 79-83). Specifically:
> > >
> > > "Although obtained images may exhibit limited natural fidelity, we believe that they contain rich semantic information necessary for model generalization, it’s just that this meaningful information is suppressed by specific artifacts and unclear semantics, thus reducing generalization."
> > >
> > > To more clearly describe the motivation behind SI, we would like to add the following emphasis:
> > >
> > > "Datasets synthesized by classical DD often exhibit ambiguous semantics, limiting their generalization. We posit that distilled datasets encapsulate rich semantic information extracted from the original data, which is merely suppressed by the ambiguity and abstraction. Therefore, we propose SI, a strategy designed to inherit the high-level semantics from distilled images. Although these images deviate from the original data distribution, our approach enhances their semantic clarity during the reverse process of a pre-trained diffusion model. Crucially, it achieves this while preserving the valuable abstract knowledge and pulling the distribution back towards the original."
> > >
> > > To avoid disrupting the index that already addresses reviewers' comments (corresponding to updated paper), we will present it in the final version.
> > >
> > > 2. We employ DDIM inversion rather than random noising over 25 steps for SI on the latent encoding $z_ 0$ of the distilled images (Images distilled using MMT under IPC10). The experimental results indicate that DDIM inversion  better preserves the initial information of $z_ 0$ compared to random noising, leading to considerable performance gains across multiple datasets.
> > >
> > > | Noise          |    Nette     |     Woof     |    Squawk    |
> > > | -------------- | :----------: | :----------: | :----------: |
> > > | Random         |   34.3±1.4   |   21.5±1.6   |   33.8±1.4   |
> > > | DDIM Inversion | **35.3±1.6** | **22.3±1.4** | **35.2±1.5** |
> > >
> > > Once again, we are grateful for your perceptive comments. We will update the final version with the new experimental results as suggested.

---

### Official Review · Reviewer_a8MY · 2025-10-29

**Soundness:** 3
**Presentation:** 3
**Contribution:** 3
**Rating:** 6
**Confidence:** 4

**Summary:**

- This paper address the issue of architecture generalization in dataset distillation
- This paper proposes a post-processing step to Dataset Distillation methods that claims to remove architecture-specific low-level details, but retain high level features which are useful to architecture generalization
- The method basically involves using the autoencoder of latent diffusion models to generate latent codes of the distilled images, then adding (partial) noise to that latent code, and then denoising using the diffusion model. The generated images retain high level patterns from the original image, but remove the low-level artifacts
- The cost of doing this method is low (several seconds per final distilled image)
- Empirically, this method outperforms GLaD-based DD methods on architecture generalization, and is compatible with any dataset distillation algorithm

**Strengths:**

- The method is well presented and the general intuition behind it is clear
- The fact that the method can be added on top of any existing DD algorithm is a major advantage
- Empirical results are strong and the increase architecture generalization is strong and consistent

**Weaknesses:**

- The paper mentions "In the specific ConvNet, we observe a notable performance degradation with our method." at the very end of the paper. This needs to be mentioned earlier in the paper, as this is a **major drawback** of the method. Indeed, Table 7 in the appendix shows a very dramatic drop in performance. I suggest discussing this in the experiments sections with table 7 moved into the main text. This tradeoff between specific architecture performance and generalization needs to be more clear
- Along the same vein as the previous point, it would be useful to show the performance of GLaD in table 7 to better understand the tradeoff of generalization performance and task-specific information

**Questions:**

- I'm struggling to understand the exact staging of the method, and the difference between SG and SF, specifically with section 3.3.3 and figure 2. If I understand correctly:
 1. Semantic Guidance (SG) uses $z_0$ generated from the original distilled image to guide the diffusion process, and is used
 2. Semantic Fusion is unclear to me - is this using the text latent embedding from the true class label (i.e. the latent encoding of the phrase "dog") to guide the diffusion? It would be useful to write out the exact equation for this.
Additionally, it would be useful in figure 2, show exactly which losses are active for each stage in the diffusion pipeline.

- The pseudocode in Algorithm 1 doesn't really mention what semantic fusion is either.

---

> ### Author Response · Authors · 2025-11-23
> **Response to Reviewer a8MY（1/1）**
>
> Thank you for your insightful questions and constructive suggestions.  We answer your questions as below.
>
> **Q1 (@Weakness 1)：The paper mentions "In the specific ConvNet, we observe a notable performance degradation with our method." at the very end of the paper. This needs to be mentioned earlier in the paper, as this is a major drawback of the method. Indeed, Table 7 in the appendix shows a very dramatic drop in performance. I suggest discussing this in the experiments sections with table 7 moved into the main text. This tradeoff between specific architecture performance and generalization needs to be more clear.**
>
> We present the data from Table 7 (Table 10 in the updated version) in Figure 3 (right) and analyze in detail the reasons for the decline in ConvNet in Section 4.4 (lines 470-482):
>
> (1) The encoder maps the distilled images into a deep latent space, filtering out architecture-specific patterns (ConvNet) while preserving informative semantics. This is evidenced by our experimental finding in Table 9 that directly decoding these latent representations (without diffusion model) still yields considerable generalization gains.
>
> (2) The diffusion model further eliminates residual architecture-specific information (degraded semantics) in the latent space through denoising steps, thereby injecting clear semantic information to enhance generalization.
>
> We have updated the results of directly **reconstructing distilled images using only VAE (without diffusion)**, which also confirms our inference (the performance gradually decreases on ConvNet).
> | Method | Architecture | Distilled | Reconstructed | Ours     |
> | ------ | :------------: | :---------: | :-------------: | :--------: |
> | DM     | ConvNet      | 23.2±0.8  | 21.2±1.2      | 19.6±0.9 |
> |        | CrossArc     | 17.6±1.7  | 18.9±1.0      | 20.0±1.6 |
> | MTT    | ConvNet      | 37.1±1.3  | 31.6±0.7      | 28.3±1.1 |
> |        | CrossArc     | 16.1±1.4  | 21.8±1.9      | 26.7±1.7 |
>
> Furthermore, we clarify the importance of generalization in Table 14 and lines 911-917.
>
> **Q2 (@Weakness 2)：Along the same vein as the previous point, it would be useful to show the performance of GLaD in table 7 to better understand the tradeoff of generalization performance and task-specific information**
>
> Because GLaD introduces GAN, it incurs significant storage and computational overhead during the synthesis stage, and is currently primarily limited to the IPC=1 setting. We supplement the results of gald with IPC=1 in Table 10 of the updated paper, and illustrate and analyze this trade-off in Figure 3 (right) and lines 478-482.
>
> Glad shows only a slight decrease in performance on ConvNet, mainly due to its one-stage architecture, which also utilizes information provided by ConvNet during distillation. However, its generalization is mediocre, and the introduction of GAN significantly increases training overhead, limiting its application. Our DIVER can be seamlessly applied to high IPC.
>
>
>
> **Q3 (@Question 1)：Semantic Guidance (SG) uses z$_0$  generated from the original distilled image to guide the diffusion process, and is used**.
>
> Yes, z$_0$  is used to guide diffusion generation in the semantic phase.
>
> **Q4 (@Question 2)：Semantic Fusion is unclear to me - is this using the text latent embedding from the true class label (i.e. the latent encoding of the phrase "dog") to guide the diffusion? It would be useful to write out the exact equation for this. Additionally, it would be useful in figure 2, show exactly which losses are active for each stage in the diffusion pipeline.**
>
> We sincerely apologize for any confusion caused regarding semantic fusion. We have updated Figure 2 and related descriptions (lines 250-258) so that you can more clearly understand each of our components.
>
> Specifically, we employ Classifier-Free Guidance (CFG) at each phase, where label information is incorporated to enhance semantic clarity (in DiT, labels are numerical (e.g., 1 for “dogs”) rather than text). Semantic fusion is essentially a "specific phase",  which is applied only during the semantic guidance phase, where we integrate the inherited semantic information from initialization, label embeddings, and the guidance signal from z$_0$. The reason for not applying fusion throughout the entire reverse process is that  z$_0$, derived from abstract distilled images, may introduce artifacts as shown in Figure 4 (right) and lead to performance degradation (as indicated in Table 7) if used for guidance across all denoising steps.
>
> We've also updated our pseudocode so you can better understand our components (lines 784-804).

---

> ### Author Response · Authors · 2025-11-27
> **A respectful request for discussion**
>
> Thanks a lot for your time in reviewing and insightful comments, according to which we have carefully revised the paper to answer the questions. We sincerely understand you’re busy. But since the discussion due is approaching, would you mind checking the response to confirm where you have any further questions?
>
> We are looking forward to your reply and happy to answer your further questions.
>
> Best regards
>
> Authors

---

### Official Review · Reviewer_xHeU · 2025-10-31

**Soundness:** 2
**Presentation:** 2
**Contribution:** 2
**Rating:** 4
**Confidence:** 4

**Summary:**

This paper proposes DIVER, a dual-stage framework to improve the poor cross-architecture generalization of classic Dataset Distillation (DD) methods. Stage I uses any standard DD algorithm to get a biased, noisy distilled dataset . Stage II, the core contribution, uses a pre-trained diffusion model (DiT) in a training-free and original-data-free manner to "refine" these images into a new, more realistic synthetic dataset. This refinement relies on three strategies: Semantic Inheritance (SI) (initializing diffusion from noisy latents) , Semantic Guidance (SG) (L2 guidance towards initial latents) , and Semantic Fusion (SF) (applying guidance only during specific timesteps). Experiments show DIVER significantly boosts cross-architecture performance for various DD methods.

**Strengths:**

1. Addresses an Important Problem: Tackles the well-known and critical issue of cross-architecture generalization in DD.

2. Highly Efficient & Practical: The method works as a "plugin" that is training-free and original-data-free, making it easy to apply.

3. Strong Empirical Gains: Consistently and significantly improves cross-architecture generalization for multiple DD baselines (Table 1, 2).

**Weaknesses:**

1. Insufficient Evaluation on Full ImageNet-1K: The paper's claims of scalability and broad applicability are undermined by a lack of thorough experimentation on the full ImageNet-1K dataset. While extensive results are provided on ImageNet subsets (e.g., Table 1, 2), the evaluation on the full 224x224 ImageNet-1K is confined to Table 3. This table only compares DIVER against two specific 'dual-time matching' methods ($SRe^{2}L$ and G-VBSM). Critically, the paper does not show how DIVER performs when combined with the other primary distillation baselines on the full dataset. This is a significant omission, as strong performance on subsets does not guarantee scalability or effectiveness in the more challenging large-scale regime.

2. Unsubstantiated Core Claim: The central hypothesis of "filtering architecture-specific noise"  is never proven and is likely a simplistic misinterpretation of a "bias replacement" effect.

3. Key Negative Result Downplayed: The method severely degrades performance on the original distillation architecture (ConvNet) , a major flaw that the authors dismiss as an "acceptable trade-off".

4. Over-marketing ("Buzzwords"): The paper relies on a "semantic salad" of invented terms (SI, SG, SF)  to inflate the perceived novelty of simple implementation steps.

5. Insufficient Ablation: The baseline for the ablation study is not minimal, making it difficult to isolate the true contribution of SI, SG, and SF beyond the inherent regularizing effect of the diffusion model itself.

**Questions:**

1. Why were the other key baselines excluded from the full ImageNet-1K evaluation presented in Table 3? To fully substantiate the claim that DIVER is a scalable and general "plug-in", it is essential to demonstrate its effectiveness on these core methods on the full dataset, not just on subsets.

2. Evidence for "Filtering": What direct evidence supports the claim of "filtering noise"  rather than simply replacing the ConvNet bias with a VAE/Transformer bias?

3. Justifying Performance Drop: How can the severe performance drop on the original ConvNet  be justified? Doesn't this indicate that DIVER is destroying useful information, not just noise?

4. Minimal Baseline: What is the cross-architecture performance of a minimal baseline using only the VAE encode/decode (i.e., $\mathcal{F}(\mathcal{E}(\tilde{x}_i))$), without any DiT denoising? This is needed to isolate the VAE's effect.

5. Guidance Design: Why guide the sampling process towards $z_0$28, which presumably contains the undesirable artifacts? Why is this better than guiding towards a cleaner target?

---

> ### Author Response · Authors · 2025-11-23
> **Response to Reviewer xHeU（1/2）**
>
> Thank you for your instructive feedback and valuable suggestions. Below are our responses to your questions.
>
> **Q1 (@Weakness 1 and Question 1)：Why were the other key baselines excluded from the full ImageNet-1K evaluation presented in Table 3? To fully substantiate the claim that DIVER is a scalable and general "plug-in", it is essential to demonstrate its effectiveness on these core methods on the full dataset, not just on subsets.**
>
> Classic DD frameworks are difficult to scale to high-resolution and large-scale datasets (e.g., 224×224 ImageNet-1K), mainly because:
>
> (1) They employ bi-level optimization to generate distilled dataset,  all performing optimization directly in pixel space is extremely time-consuming.
>
> (2) Training multiple expert trajectory models on large datasets to build a buffer is also very time-consuming.
>
> (3) The optimization process involves storing and computing the gradient graph of multiple gradient descent steps, which requires hundreds of Gigabytes of memory and a significant amount of training time.
>
> Therefore, classic DD methods are **generally limited** to the  evaluation of ImageNet subset. **None of the recent studies on classical DD (such as NCFM(CVPR 2025) [1], EDF (CVPR 2025)[2], HDD (NeurIPS 2025) [3], etc.) have been able to successfully scale to the entire Imagenet-1K dataset**.
>
> In addition to the "dual-time matching" method, we have supplemented the updated paper with experiments (Figure 4) and analysis (lines 366–368) on ImageNet-1K based on the diffusion method. Our method can reliably improve MGD$^{3}$ on large-scale ImageNet-1K. Furthermore, after fine-tuning with Minimax, we achieve further performance improvement (achieve SOTA), demonstrating the superior scalability.
>
> | IPC  |  SRe²L  | G-VBSM   | RDED     | DiT      | Minimax  | D⁴M      | MGD³     | MGD³+Ours | MGD³+Minimax+Ours |
> | :--: | :------: | :------: | :------: | :------: | :------: | :------: | :------: | :-------: | :---------------: |
> | 10   | 21.3±0.6 | 31.4±0.5 | 42.0±0.1 | 39.6±0.4 | 44.3±0.5 | 27.9±0.2 | 45.8±0.3 | 46.4±0.3  | **46.9±0.5**          |
> | 50   | 46.8±0.2 | 51.8±0.4 | 56.5±0.1 | 52.9±0.6 | 58.6±0.3 | 55.2±0.1 | 60.2±0.1 | 60.6±0.4  | **61.0±0.2**          |
>
> **Q2 (@Weakness 2 and Question 2)：Evidence for "Filtering": What direct evidence supports the claim of "filtering noise" rather than simply replacing the ConvNet bias with a VAE/Transformer bias?**
>
>  From the results in Figure 7 of the appendix, we notice that the images reconstructed by VAE (direct encoding and decoding without DiT) are "clearer" than those in the distilled dataset. This qualitatively confirms that encoding high-dimensional images into low-dimensional images by VAE does indeed filter out "noise." On the other hand, the experimental results in Table 9 show that the distillation performance of the reconstructed images on ConvNet decreases, which quantitatively confirms that "noise" is filtered.
>
> **Q3 (@Weakness 3 and Question 3)：Justifying Performance Drop: How can the severe performance drop on the original ConvNet be justified? Doesn't this indicate that DIVER is destroying useful information, not just noise?**
>
>  We attribute the poor generalization of the distilled dataset to **noise coverage** and **semantic degradation**. This phenomenon is caused by overfitting to a specific pattern of ConvNet. Therefore, when the "noise" and "unclear semantics" derived from ConvNet are removed, the performance degradation of ConvNet is inevitable. However, the **improved generalization performance and enhanced robustnes** (Table 14, lines 911-917) under different settings confirm that **useful information** is preserved.
>
> Furthermore, we respectfully emphasize that generalization performance is more critical than distillation performance on ConvNet, since in practical applications, the distilled dataset is typically used to train or fine-tune other architectures rather than being applied to a simple ConvNet with only 3–6 convolutional layers.
>
> **Q4 (@Weakness 4)：Over-marketing ("Buzzwords"): The paper relies on a "semantic salad" of invented terms (SI, SG, SF) to inflate the perceived novelty of simple implementation steps.**
>
> We fully understand the confusion you may have regarding the new terms (SI, SG, SF) introduced in the paper, and we apologize for our failure to clearly explain their necessity and substance. We have revised Figure 2 and its related descriptions (lines 250-258) to more clearly describe each of our components. Undeniably, our method is **simple yet effective**, and it is precisely this simplicity that makes it highly **efficient** (requiring processing time comparable to raw DiT on ImageNet (256 X 256) with only 4.02 GB of GPU memory usage).
>
>
>
>
> [1] Dataset Distillation with Neural Characteristic Function: A Minmax Perspective, CVPR 2025.
>
> [2] Emphasizing Discriminative Features for Dataset Distillation in Complex Scenarios, CVPR 2025.
>
> [3] Hyperbolic Dataset Distillation, NeurIPS 2025.

---

> ### Author Response · Authors · 2025-11-23
> **Response to Reviewer xHeU（2/2）**
>
> **Q5 (@Weakness 5)：Insufficient Ablation: The baseline for the ablation study is not minimal, making it difficult to isolate the true contribution of SI, SG, and SF beyond the inherent regularizing effect of the diffusion model itself.**
>
> The results in Table 7 demonstrate the positive contribution of each component. To rule out the regularization effect of Diffusion, we also apply **randomly selected images (Random$^{*}$ in Table 7) and diffusion-based synthesized images (Synthesis-based in Table 11)** in our framework, but observe no significant performance improvement or even a performance drop (51.7% vs. 60.3%). This confirms that each of our components effectively extracts valuable information from the distilled dataset.
>
> **Q6 (@Question 4)：Minimal Baseline: What is the cross-architecture performance of a minimal baseline using only the VAE encode/decode, without any DiT denoising? This is needed to isolate the VAE's effect.**
>
> Thank you for raising this important question, as it has given us a deeper understanding of each component of DIVER. We have supplemented the revised version of our paper with experiments that employ only the VAE (Table 9, lines 459-467).
> We first visualize in Figure 7 the images reconstructed by the VAE (without diffusion), which appear “cleaner’’ than the distilled dataset, providing evidence that SI can effectively filter "noise". As shown in Table 9, the reconstructed images also enhance generalization to some extent. Furthermore, after semantic refinement with diffusion (Ours), the generalization performance improves even more. Combined with the performance drop observed on ConvNet, these results support our view that poor generalization is caused by noise coverage and semantic degradation.
>
> | Method | Architecture | Distilled | Reconstructed | Ours     |
> | ------ | :------------: | :---------: | :-------------: | :--------: |
> | DM     | ConvNet      | 23.2±0.8  | 21.2±1.2      | 19.6±0.9 |
> |        | CrossArc     | 17.6±1.7  | 18.9±1.0      | 20.0±1.6 |
> | MTT    | ConvNet      | 37.1±1.3  | 31.6±0.7      | 28.3±1.1 |
> |        | CrossArc     | 16.1±1.4  | 21.8±1.9      | 26.7±1.7 |
>
> **Q7 (@Question 5)：Guidance Design: Why guide the sampling process towards 28, which presumably contains the undesirable artifacts? Why is this better than guiding towards a cleaner target?**
>
> Since the objective of diffusion is to transform a Gaussian distribution into a data distribution, the latent code z$_0$ encoded from distilled images does not follow a Gaussian distribution. Therefore, **we add 25 steps of noise to z$_0$ to initialize the latent, which preserves the original information while conforming to a Gaussian distribution.** Excessive noise addition would cause information loss, and the results in Figure 3 (Medium) demonstrate that adding an appropriate number of noise steps yields the best performance. **During the semantic guidance phase, we still use the clean z$_0$  rather than the noised z$_0$.**

---

> ### Author Response · Authors · 2025-11-27
> **A respectful request for discussion**
>
> Dear Reviewer,
>
> Thanks a lot for your time in reviewing and insightful comments, according to which we have carefully revised the paper to answer the questions. We sincerely understand you’re busy. But since the discussion due is approaching, would you mind checking the response to confirm where you have any further questions?
>
> We are looking forward to your reply and happy to answer your further questions.
>
> Best regards
>
> Authors

---

### Author Response · Authors · 2025-11-27
**Overview of Revisions in the Resubmitted Manuscript based on Reviewer Feedback**

We sincerely appreciate the reviewers for their valuable and insightful suggestions as well as their recognition of our work, particularly their positive comments on its **"innovation", "efficiency", "practicality", "clear motivation" and "strong and consistent empirical gains"**. Our revisions, detailed in the point-by-point responses, have diligently incorporated their feedback to enhance the study. Below is a summary of the major changes.

1. We provide a detailed explanation of why conventional DD methods struggle to scale to high-resolutions and large-scale datasets , and we further supplement our study with diffusion-based experiments on **ImageNet-1K in Table 4**.（Q1@Reviewer xHeU, Q3@Reviewer QTpw)

2. We include the ablation study  in **Table 9** and visualizations in **Figure 7**  using **only the VAE** to further substantiate our claim regarding its ability to **filter out “noise”**.（Q2@Reviewer xHeU, Q1@Reviewer QTpw, Q6@Reviewer D6Z8)

3. In **Fig. 2** and the description in **lines 250–258**, we **refine the activity status of each component** (particularly SF) throughout the diffusion process to improve the clarity of the presentation. (Q4@Reviewer a8MY, Q2@Reviewer QTpw)

4. We supplement **Table 10** with the performance of GLaD on ConvNet, and illustrate it in **Figure 3 (right)**. We then analyze this trade-off (lines 470-482) and **emphasize the importance of generalization** (lines 911-917). (Q2@Reviewer a8MY, Q3@Reviewer xHeU)

5. We provide a detailed explanation of why diffusion-based methods **do not directly use the synthesized real images but instead optimize the prototypes**, and **Table 11** further validates the degradation caused by using real images. (Q5@Reviewer QTpw, Q1@Reviewer D6Z8)

6. We supplement **Table 8** with experiments assessing DIVER’s **sensitivity to different diffusion** settings.  (Q4@Reviewer QTpw)

7. In **Section A.2.3** of the Related Work in our updated paper, we analyze the **connection between diffusion-based image restoration and our DIVER**. (Q3@Reviewer D6Z8)

8. In **Section 2.1 and Section 3.1**, we completely revise the presentation to conform to the **typical formulation of dataset distillation**. (Q2@Reviewer D6Z8)

---

### Meta-Review · Area_Chair_tyDD · 2025-12-17

**Summary:**

Reviewer xHeU pointed out the missing evaluation on full ImageNet-1K for the validation of scalability, negative results on original architecutre (architecture-specific noise) and insufficient ablation experiments. Reviewer a8MY also found that the proposed method suffers a very dramatic performance drop on specific ConvNet. Reviewer QTpw focused on the mathematical analysis of SI, SG and SF, then mentioned that the results don't obviously outperform sota diffusion-based methods. Reviewer D6Z8 argued that the explanation about performance gain over realistic distilled images.

**Reviewer Concerns:**

The explanation about how the proposed method filter out architecture-specific noise raised by Reviewer xHeU and Reviewer QTpw is not clearly clarified. The authors claimed that VAE latent can filtered "noise" due to its semantic representation. This explanation is not well sound enough, where VAE is also based on ConvNet and focues on low-level features.

**Reviewer Scores:**

Reviewer xHeU: 4;

Reviewer a8MY: 6;

Reviewer QTpw: 4;

Reviewer D6Z8: 4;

---

### Decision · Program_Chairs · 2026-01-26

Reject